# Visualizing protein breathing motions associated with aromatic ring flipping

Laura Mariño Pérez[1,2], Francesco S. Ielasi[3], Luiza M. Bessa[1], Damien Maurin[1], Jaka Kragelj[1,5], Martin Blackledge[1], Nicola Salvi[1], Guillaume Bouvignies[4], Andrés Palencia[3 ✉] & Malene Ringkjøbing Jensen[1 ✉]

Aromatic residues cluster in the core of folded proteins, where they stabilize the structure through multiple interactions. Nuclear magnetic resonance (NMR) studies in the 1970s showed that aromatic side chains can undergo ring flips—that is, 180° rotations—despite their role in maintaining the protein fold[1–3]. It was suggested that large-scale 'breathing' motions of the surrounding protein environment would be necessary to accommodate these ring flipping events[1]. However, the structural details of these motions have remained unclear. Here we uncover the structural rearrangements that accompany ring flipping of a buried tyrosine residue in an SH3 domain. Using NMR, we show that the tyrosine side chain flips to a low-populated, minor state and, through a proteome-wide sequence analysis, we design mutants that stabilize this state, which allows us to capture its high-resolution structure by X-ray crystallography. A void volume is generated around the tyrosine ring during the structural transition between the major and minor state, and this allows fast flipping to take place. Our results provide structural insights into the protein breathing motions that are associated with ring flipping. More generally, our study has implications for protein design and structure prediction by showing how the local protein environment influences amino acid side chain conformations and vice versa.

Aromatic residues make up a considerable fraction of the hydrophobic core of folded proteins, where they stabilize the structure through CH–π (refs. [4–6]), π–π (refs. [7,8]) and cation–π (refs. [9,10]) interactions as well as hydrogen bonds. NMR studies in the 1970s demonstrated that aromatic side chains can undergo ring flips—that is, 180° rotations of the $\chi_2$ dihedral angle (Cβ–Cγ axis)—even when engaged in stabilizing interactions in the hydrophobic core[1–3]. These ring flips require concerted movements of the surrounding residues (large-amplitude protein breathing motions), and ring flipping rates as a function of temperature and pressure have been used to report on these motions by deriving activation energies and volumes[11–21]. However, the structural details of ring flipping and the associated breathing motions have remained unknown, probably owing to difficulties in stabilizing ring flipping transition states or intermediates that are amenable to structure elucidation.

Here we capture ring flipping events of a buried tyrosine residue in the SH3 domain of the JNK-interacting protein 1 (JIP1). We show using NMR relaxation dispersion that the aromatic ring of this tyrosine residue populates a minor-state conformation (3%), and we design single point mutations to stabilize this conformation and capture its high-resolution structure using X-ray crystallography. The structure reveals how the intricate network of hydrogen bonds and CH–π interactions is rearranged in the minor state. We show how a substantial void volume is generated around the tyrosine

ring during the structural transition from the major to the minor state, which can be associated with the breathing motions that allow fast-timescale ring flipping events to take place. Our results provide structural insights into aromatic ring flipping and its associated protein breathing motions.

## Protein dynamics induced by a tyrosine residue

The SH3 domain of JIP1 undergoes exchange between two distinct conformations, as evidenced by $^{15}$N NMR relaxation measured at multiple temperatures (Extended Data Figs. 1, 2, Supplementary Discussion). To analyse the observed dynamics in detail, we acquired $^{15}$N and $^1$H$^N$ Carr–Purcell–Meiboom–Gill (CPMG) relaxation dispersion experiments at 15 °C (refs. [22–25]). These experiments quantify the kinetics of exchange processes and provide the difference in chemical shift between a major and a minor state, together with their relative populations[26–30]. The data confirm that exchange contributions to the transverse relaxation are present for residues within three regions of the protein (Extended Data Fig. 1d, e, Supplementary Table 1). These residues are located spatially close to tyrosine 526 (Y526) (Extended Data Fig. 1f). A mutation of Y526 to alanine (Y526A) shows no conformational exchange (Extended Data Fig. 1d, e), and conserves the protein backbone conformation, as evidenced from its crystal structure that we obtained at 1.5-Å resolution (Extended

[1]Université Grenoble Alpes, CEA, CNRS, IBS, Grenoble, France. [2]Departament de Química, Universitat de les Illes Balears, Palma de Mallorca, Spain. [3]Institute for Advanced Biosciences (IAB), Structural Biology of Novel Targets in Human Diseases, INSERM U1209, CNRS UMR5309, Université Grenoble Alpes, Grenoble, France. [4]Laboratoire des Biomolécules (LBM), Département de Chimie, École normale supérieure, PSL University, Sorbonne Université, CNRS, Paris, France. [5]Present address: Department of Biophysics, University of Texas Southwestern Medical Center, Dallas, TX, USA. ✉e-mail: andres.palencia@inserm.fr; malene.ringkjobing-jensen@ibs.fr

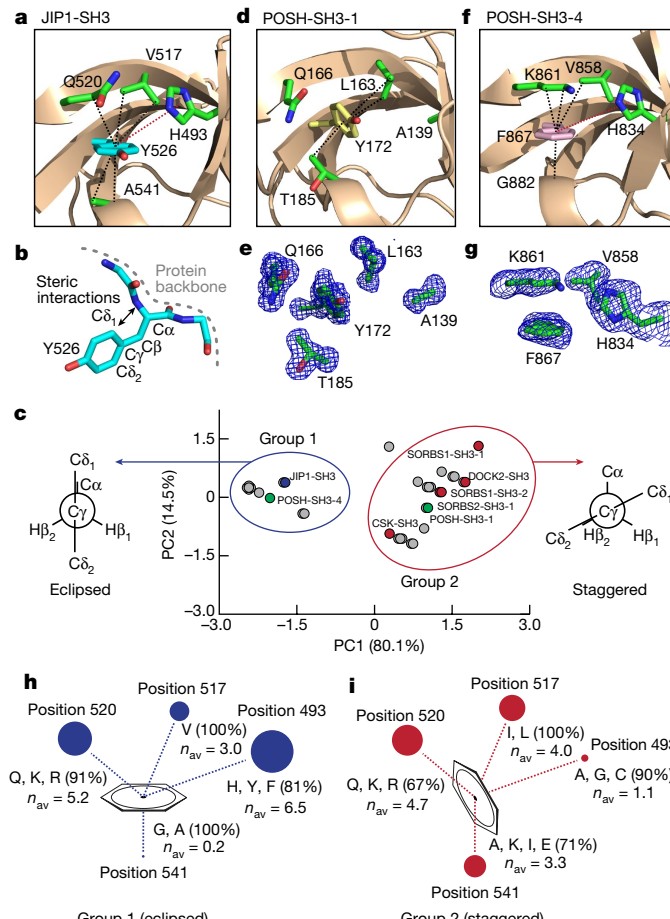

**Fig. 1 | The side chain conformation of Y526 is determined by steric interactions with the surrounding amino acids. a**, Crystal structure of wild-type JIP1-SH3, showing the conformation of Y526 and its stabilizing interactions with H493, V517, Q520 and A541. Dashed lines indicate CH–π (black) and π–π (red) interactions. **b**, Side chain conformation of Y526 in JIP1-SH3, illustrating the steric interactions between the $\delta_1$ nuclei of the aromatic ring and the backbone. **c**, PCA of a dataset comprising the size of the amino acid side chains at positions 493, 517 and 541 within SH3 domains that contain Y or F at position 526 (Extended Data Fig. 5b). Two groups are observed, which correspond to eclipsed (group 1) or staggered (group 2) conformations of the aromatic ring. JIP1-SH3 is indicated in blue; POSH-SH3-1 and POSH-SH3-4 are indicated in green; and SH3 domains for which crystal structures have been determined previously are shown in red. **d**, Crystal structure of POSH-SH3-1, showing a staggered conformation of Y172. **e**, Unbiased electron density maps (Fo–Fc) of Y172 and the surrounding residues in POSH-SH3-1. **f**, Crystal structure of POSH-SH3-4, showing an eclipsed conformation of F867. **g**, Unbiased electron density maps (Fo–Fc) of F867 and the surrounding residues in POSH-SH3-4. **h, i**, Results of the PCA, illustrating the size and nature of the residues that surround the aromatic ring in position 526 in group 1 (**h**) and group 2 (**i**) SH3 domains. The size of the spheres in each position is proportional to the average size ($n_{av}$) of the amino acid side chain across group members.

Data Fig. 3, Extended Data Tables 1, 2). These results show that the relaxation dispersion that affects around 40% of the residues in the SH3 domain arises from a single exchange process, with Y526 being the origin of the observed exchange.

The $^{15}$N and $^{1}$H$^{N}$ relaxation dispersion data were analysed simultaneously according to a two-site exchange model in which a highly populated major state interconverts with a low-populated minor state (Extended Data Fig. 4a, b, Supplementary Discussion). The analysis of the data gives a population of the minor state of $p_{minor} = 2.8 \pm 0.1\%$ and

an exchange rate constant of $k_{EX} = 2,600 \pm 70$ s$^{-1}$. The derived chemical shift differences, $\Delta\delta_{CPMG}$, span a range of 4.7 ppm for $^{15}$N and 1.1 ppm for $^{1}$H$^{N}$ suggesting that there are substantial structural changes between the major and the minor state (Extended Data Fig. 4c, d).

The side chain of Y526 is found in an unusual conformation in the crystal structure (Protein Data Bank (PDB) code 2FPE (ref. [31]), Extended Data Table 1), characterized by a $\chi_2$ dihedral angle of 2° (Fig. 1a). Normally, this eclipsed conformation is energetically unfavourable because of steric interactions with the backbone (Fig. 1b), and it is rarely found in proteins as $\chi_2$ angles are preferred where C$\delta_1$ and C$\delta_2$ are staggered with respect to C$\alpha$ (ref. [32]). However, the eclipsed conformation of the aromatic ring of Y526 is stabilized by CH–π interactions from V517, Q520 and A541 and by π–π interactions with H493 (Fig. 1a).

## Proteome-wide SH3 sequence analysis

To investigate the contribution from the surrounding residues in stabilizing the eclipsed conformation of Y526, we analysed the sequences of all identified human SH3 domains[33]. We categorized the sequences according to the identity of the amino acid at the position of Y526 in the JIP1 SH3 domain (JIP1-SH3) (Extended Data Fig. 5a), and we retained the sequences carrying a phenyl-based amino acid (Tyr or Phe) at this position, amounting to 33 SH3 domains. Sequence alignments reveal a large variation in the size of the amino acids at positions 493, 517 and 541, whereas at position 520 most sequences contain Gln, Arg or Lys (72%) (Extended Data Fig. 5b, c). To study the size correlation between the amino acids at positions 493, 517 and 541 and their influence on the conformation of the aromatic residue at position 526, we carried out a principal component analysis (PCA) by assigning a size score ($n$) to each amino acid according to the number of heavy atoms in their side chains. This analysis reveals two well-separated groups, with the SH3 domain of JIP1 belonging to group 1 (Fig. 1c). In group 2, five SH3 domains are found for which high-resolution crystal structures are available; these include three SH3 domains of the sorbin and SH3 domain-containing proteins 1 and 2 (SORBS1 and SORBS2)[34], and the SH3 domains of the dedicator of cytokinesis protein 2 (DOCK2)[35] and of the tyrosine protein kinase CSK[36] (Extended Data Fig. 5b). Notably, all group 2 structures show a favourable, staggered side chain conformation (of C$\delta_1$/C$\delta_2$ with respect to C$\alpha$) of the corresponding tyrosine, with the $\chi_2$ dihedral angle ranging from −40° to −64° (Fig. 1c). We therefore hypothesized that SH3 domains of group 1 have eclipsed conformations, whereas group 2 have staggered conformations. To test our hypothesis, we determined two crystal structures of SH3 domains of the scaffold protein POSH ('plenty of SH3 domains'), for which the first SH3 domain belongs to group 2 and the fourth SH3 domain belongs to group 1 (Extended Data Table 2, Fig. 1c). Consistent with the PCA, the crystal structure of the first SH3 domain of POSH (POSH-SH3-1) shows a staggered conformation of the corresponding tyrosine residue (Y172), which is stabilized by CH–π interactions from L163 (position 517) and T185 (position 541) (Fig. 1d, e). POSH-SH3-4 shows a similar structure to JIP1-SH3, with an eclipsed conformation of the corresponding phenylalanine residue (F867) stabilized by CH–π interactions from V858 (position 517), K861 (position 520) and G882 (position 541), and by π–π interactions with H834 (position 493) (Fig. 1f, g). Our data therefore suggest that the conformation of the aromatic ring at position 526 is determined by steric interactions dictated by the size of the surrounding amino acids.

## Structure of the minor state

Next, we investigated whether the minor state detected by NMR corresponds to a staggered conformation of the side chain of Y526. We sought to stabilize the minor state relative to the major state by

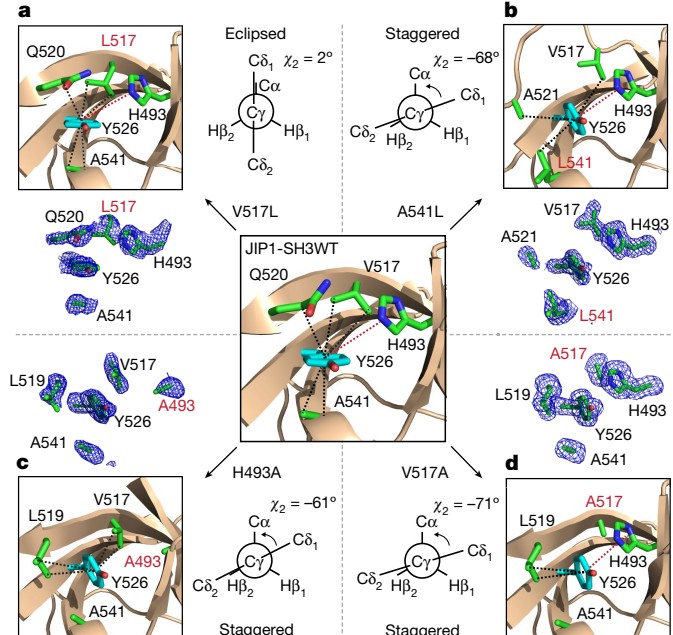

**Fig. 2 | High-resolution crystal structures of JIP1-SH3 variants. a–d**, Crystal structures showing the conformation of Y526, the corresponding unbiased electron density maps (Fo–Fc) of Y526 and its surrounding residues, and the Newman projection along the Cβ–Cγ bond of Y526 in the V517L (**a**), A541L (**b**), H493A (**c**) and V517A (**d**) variants of JIP1-SH3. Dashed lines indicate CH–π (black) and π–π (red) interactions. Residue numbers in red indicate the site of mutation. The wild-type JIP1-SH3 structure is shown as a reference in the centre.

introducing single point mutations. The PCA suggests that a staggered conformation in group 2 is favoured over an eclipsed conformation in group 1 when residues with larger side chains are found in position 541, when isoleucine or leucine are occupying position 517 and when smaller residues are found in position 493 (Fig. 1h, i). Accordingly, we designed four different mutants of JIP1-SH3 (H493A, V517A, V517L and A541L) and obtained their high-resolution structures by X-ray crystallography (Extended Data Tables 1, 2). Of note, three mutants (H493A, V517A and A541L) induce a staggered conformation of Y526, with $\chi_2$ dihedral angles ranging from −41° to −75°, whereas V517L shows an eclipsed conformation of Y526 and an almost identical structure to the wild-type protein (Fig. 2, Extended Data Fig. 6a). The high resolution of these structures, ranging from 1.4 to 1.9 Å, allows unambiguous determination of the conformation of Y526 and the surrounding residues, as demonstrated by their unbiased electron density maps (Fig. 2).

## Structural details of breathing motions

The two variants H493A and V517A show almost identical crystal structures (Extended Data Fig. 6b) and an equivalent stabilization mechanism of the aromatic ring of Y526. Whereas the wild-type structure exhibits a classic β-bulge at residue 518 (ref. [37]), the transition from the eclipsed to the staggered conformation induces a local inversion (in-out) at residues 518–520, which leads to the formation of a canonical β-strand, as observed in the structures of the H493A and V517A variants (Fig. 3a–c). This transition allows the side chain of L519 to rearrange and form CH–π interactions with the ring of Y526 (Fig. 2c, d); and, at the same time, large-scale movements of E518, Q520, E522 and Y524 are observed (Fig. 3a). We note that SH3 domains in both group 1 and group 2 of the PCA show classic β-bulges at position 518 (Extended Data Fig. 5d), which suggests that the presence of this structural motif

is not determinant of the side chain conformation of the phenyl ring in position 526.

The A541L mutation also triggers a staggered conformation of Y526 and a rearrangement of the 517–522 region; however, the stabilization mechanism of the aromatic ring is different. A looping out of the β-strand between residues 517 and 522 is observed (Extended Data Fig. 6c–e), which allows the side chain of A521 to reorient and to stabilize the staggered conformation of the ring through CH–π interactions together with L541 and V517 (Fig. 2b). At the same time, the side chains of L519, Q520 and E522 and D523 undergo large-scale movements to accommodate the flipped ring within the pocket (Extended Data Fig. 6c).

Altogether, the different mutants show that the dynamics of the region encompassing residues 517–522 are key for the formation of the minor state. The experimental [13]C chemical shifts for residues in this region are characteristic of random coil conformations (Extended Data Fig. 1b) and, compared to other regions of secondary structure, the relaxation-derived order parameters ($S^2$) are lower (Extended Data Fig. 2g) and the crystallographic $B$-factors are higher. This supports the idea of the 517–522 region being intrinsically dynamic, with a fluctuating hydrogen-bonding network that is prone to structural transitions.

Next, we sought to determine which of the two crystal structures (H493A/V517A-like or A541L-like) best capture the conformation of the wild-type minor state detected by NMR relaxation dispersion. The H493A and V517A variants show almost identical crystal structures and for a subset of residues, the chemical shifts of which are unaffected by the mutations, the resonances of the two variants fall on a straight line together with the resonances of the wild-type protein (Fig. 3d, e). This suggests that they are in fast–intermediate exchange between two conformations represented by the crystal structures of the wild-type protein and of the H493A/V517A variants. In agreement with this, both variants show line broadening and chemical exchange contributions as detected by [15]N and [1]H[N] relaxation dispersion (Extended Data Figs. 7a–c, 8a–c). Analysis of these data (Extended Data Figs. 7d, e, 8d, e) shows that the structural features of the minor state of the wild-type protein are captured by the H493A/V517A crystal structures, as shown by the excellent agreement between the $\Delta\delta_{CPMG}$ values for the wild-type protein and the two variants (Extended Data Figs. 7f, g, 8f, g). In addition, the analysis yields exchange rates between the staggered (canonical β-strand) and the eclipsed (classic β-bulge) conformation of $k_{EX} = 2,830 \pm 70$ s$^{-1}$ (H493A) and $k_{EX} = 6,800 \pm 300$ s$^{-1}$ (V517A), compared to $k_{EX} = 2,600 \pm 70$ s$^{-1}$ determined for the wild-type protein. Finally, the observable chemical shifts of the two variants (Fig. 3d, e), in conjunction with analysis of the relaxation dispersion data (Supplementary Discussion), show that the H493A mutation slightly stabilizes the minor state relatively to the major state, whereas the V517A mutation almost inverts the relative populations of the major and minor states (Fig. 3f). For completeness, we note that the A541L crystal structure is not representative of the minor state conformation (Extended Data Fig. 9), although it shares structural features that are necessary for fast ring flipping of Y526 (see below).

## Void volume enables ring flipping

Aromatic [1]H–[13]C heteronuclear single quantum coherence (HSQC) spectra show averaging of the NMR signals of the tyrosine $\varepsilon_1/\varepsilon_2$ nuclei of Y526 (Fig. 4a, Extended Data Fig. 10a, Supplementary Fig. 1), which shows that full 180° ring flipping occurs. This poses the question of the timescale of the full ring flipping and its relation to the observed minor state. To answer this question, we acquired L-optimized TROSY-selected aromatic side chain [13]Cε CPMG (ref. [38]) and on-resonance $R_{1\rho}$ (ref. [39]) relaxation dispersion data of Y526. These data are entirely explained by the exchange process between the major and minor state, with a negligible contribution from the

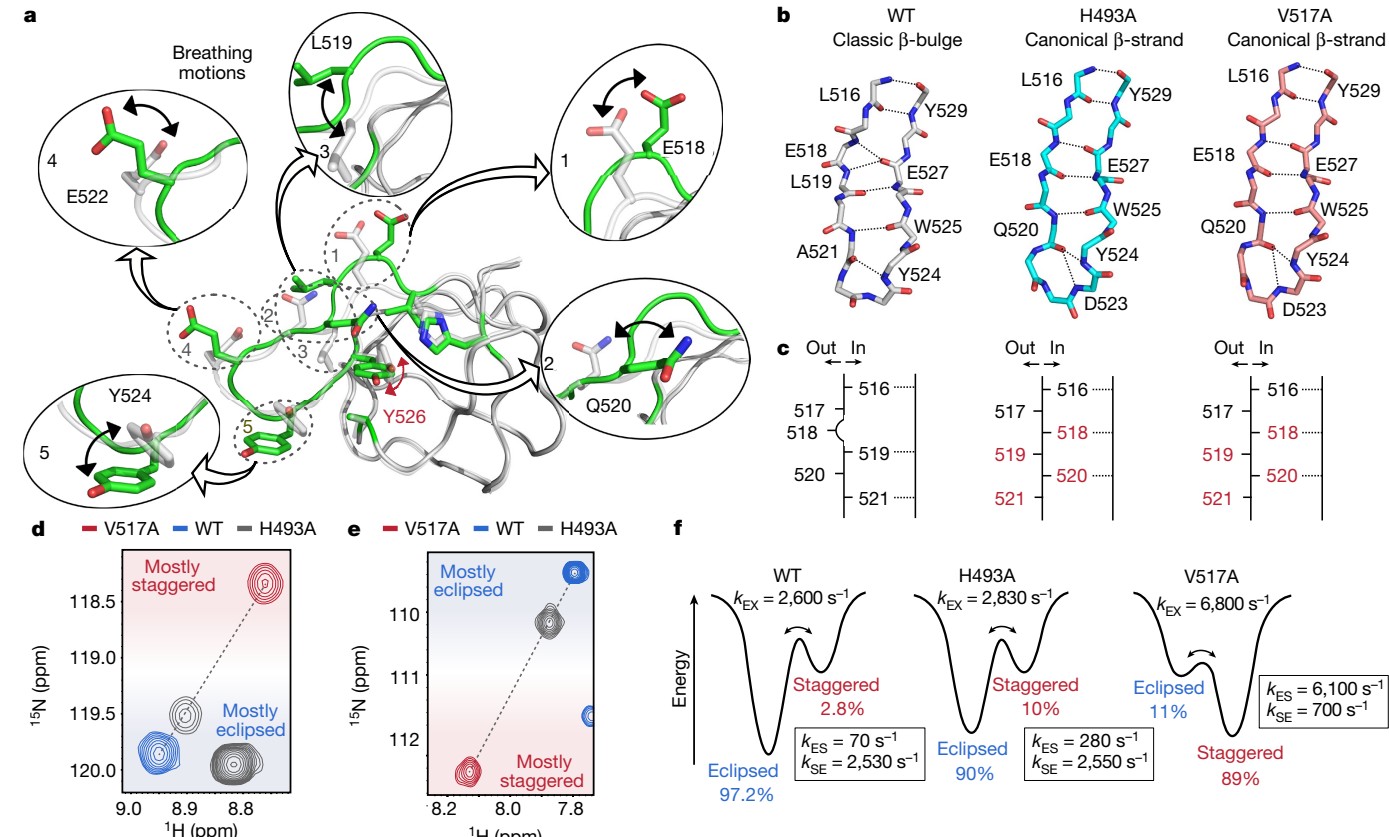

**Fig. 3 | Crystal structures capture large-scale protein breathing motions.** **a**, Structural changes associated with the transition from the eclipsed (green) to the staggered (grey) conformation of Y526. **b**, Illustration of the backbone conformation of the β-sheet formed between the 516–521 and 524–529 regions in the wild-type (WT) protein (left), and in the H493A (middle) and V517A (right) variants. Dashed lines indicate hydrogen bonds. **c**, Schematic representation of the conformation of the β-strand encompassing residues 516–521, showing the orientation of the carbonyl group ('out', carbonyl group surface exposed; 'in', carbonyl group pointing towards the β-strand encompassing residues 524–529) in the wild-type protein (left) and in the H493A (middle) and V517A (right) variants. **d**, **e**, Two examples of linear correlations between the chemical shifts of wild-type JIP1-SH3 (blue spectrum) and the two variants H493A (grey spectrum) and V517A (red spectrum) as observed in $^1$H–$^{15}$N HSQC spectra acquired at 35 °C (**d**, residue E522; **e**, residue D524). **f**, Energy landscapes illustrating the effect of single point mutations on the exchange rate constants and fractional populations of the major (eclipsed) and minor (staggered) conformations as determined by relaxation dispersion experiments acquired at 15 °C.

full ring flipping event (Fig. 4b, c), demonstrating that ring flipping of Y526 is fast ($k_{EX} > 50,000$ s$^{-1}$) (Supplementary Discussion). This observation agrees with a 1-μs molecular dynamics (MD) simulation that shows several 180° ring flipping events of Y526 (Fig. 4d, Extended Data Fig. 10b, c).

During the structural transition between the major and the minor state (Supplementary Video 1), a void volume is created around the ring of Y526 that corresponds to a pocket expansion of 65 Å$^3$; this is mainly due to the structural reorganization of the side chain of Q520 (Fig. 4e, f). This cavity expansion is in agreement with previous studies that have reported activation volumes between 40 and 85 Å$^3$ for ring flipping events of aromatic residues in other proteins[12,14,21,40,41]. The expansion is followed by a compaction of the surrounding protein environment as the ring becomes stabilized by CH–π interactions from L519 (Fig. 4e, f).

Collectively, our results are consistent with a model in which fast protein breathing motions along the structural trajectory between the major and the minor state generate the necessary void volume for ring flipping to take place by lowering the energy of the transition state (Fig. 4e–g, Supplementary Video 2). Occasionally, the β-bulge to β-sheet transition is completed and the aromatic ring becomes trapped in a staggered conformation that is stabilized by CH–π interactions with L519–a process that gives rise to the observed relaxation dispersion. These events are rare and occur on a slow timescale (Figs. 3f, 4g), but they constitute an important tool for observing the trajectory of the protein breathing motions coupled to aromatic ring flipping. The initial generation of void volume around the ring is almost identical along the structural trajectory between the major state (wild type) and the A541L crystal structure (Extended Data Fig. 6f). Thus, all mutants—including A541L, which stabilizes Y526 in a staggered conformation by a different mechanism—share the same initial structural trajectory and report on identical breathing motions.

## Conclusions

Our results provide structural insights into the protein breathing motions that are associated with aromatic ring flipping. We reveal how the dynamics of the region encompassing residues 517 to 522 are key for accommodating the ring flipping process of Y526. Notably, the transition from the eclipsed, major conformation to the staggered, minor conformation is associated with a structural change from a rare, classic β-bulge to a common, canonical β-strand conformation (Supplementary Video 1, Fig. 3a–c). Breathing motions along the structural trajectory between the major and the minor state generate the necessary void volume for fast ring flipping of Y526 to take place (Supplementary Video 2). Although a recent NMR study suggested extensive local unfolding as the source of cavity creation[41], our study provides an alternative view by showing how a substantial void volume can be

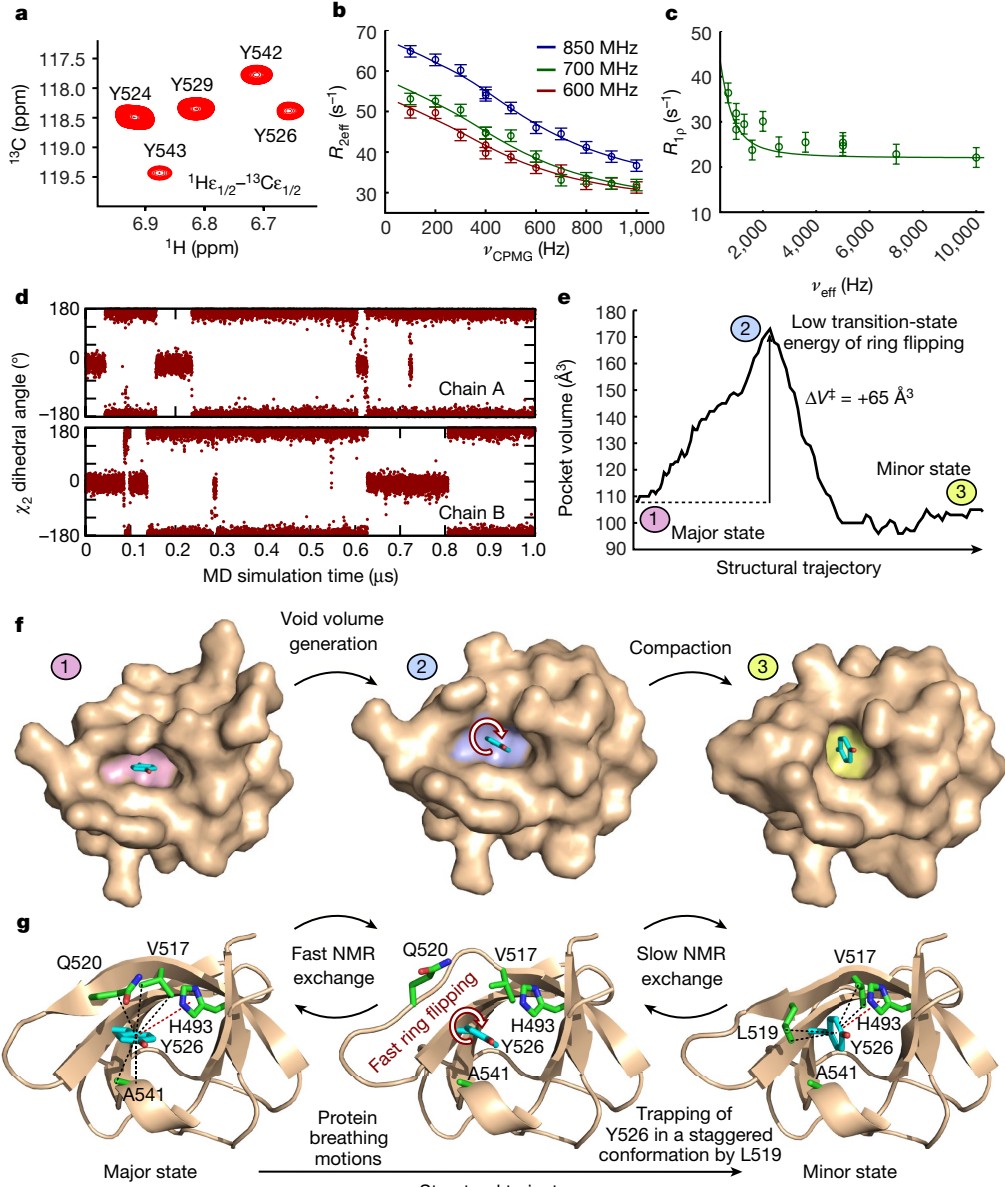

**Fig. 4 | Void volume creation enables fast aromatic ring flipping of Y526.**
**a**, $^1H$–$^{13}C$ HSQC spectrum of JIP1-SH3 at 15 °C showing tyrosine epsilon correlations. **b**, $^{13}C\epsilon$ CPMG relaxation dispersion profiles of Y526 obtained at 15 °C (red, 600 MHz; green, 700 MHz; blue, 850 MHz). The data were analysed simultaneously according to a two-site exchange model (full-drawn lines). Error bars represent one s.d. derived from Monte Carlo simulations of experimental uncertainty. **c**, $^{13}C\epsilon$ on-resonance $R_{1\rho}$ relaxation dispersion profile of Y526 at 15 °C and 700 MHz. The full-drawn line corresponds to the calculated $R_{1\rho}$ profile from the exchange parameters ($k_{EX}$, $p_{minor}$ and $\Delta\delta_{CPMG}$) obtained from the analysis of the CPMG data in **b**. Error bars represent one s.d. and were derived as in **b**. **d**, The dihedral angle $\chi_2$ of Y526 as a function of simulation time in a 1-μs MD simulation of the dimeric JIP1-SH3. **e**, Volume of the pocket of Y526 across the structural trajectory between the major and minor

state. **f**, Surface representation of JIP1-SH3 in three different states corresponding to the major state, an intermediate state on the structural trajectory and the minor state. The Y526 pocket is highlighted in pink (major), blue (intermediate) and yellow (minor). The rearrangements along the structural trajectory between the major and the minor state generate a void volume around Y526, thereby lowering the transition-state energy of ring flipping. **g**, Illustration of the protein breathing motions along the structural trajectory from the major to the minor state. A void volume is created around Y526, which allows fast ring flipping to take place. The ring flipping is occasionally interrupted by trapping of Y526 in a staggered conformation through formation of CH–π interactions with L519 enabled by the β-bulge to β-sheet transition.

generated through distinct structural rearrangements, while maintaining the overall protein fold.

More generally, our study shows how the local environment in the protein core can lead to a priori energetically unfavourable conformations of amino acid side chains, and how subtle changes in this environment can lead to major structural rearrangements to revert to preferred amino acid conformations. Our results therefore have implications for protein design and structure prediction, and for

how novel biological functions can be acquired during the course of evolution; for example, by altering the delicate balance between hydrogen bonds and CH–π interactions. Finally, the combination of sensitive NMR methods to detect low-populated states, protein design using proteome-wide sequence analyses and high-resolution crystallography could be a strategy to further discover the structural details of sparsely populated protein states and their link to function.

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

# Methods

## Expression and purification of JIP1-SH3, POSH-SH3-1 and POSH-SH3-4

JIP1-SH3, corresponding to residues 490–549 of human JIP1 (Uniprot Q9UQF2), was subcloned into a pET28a vector, and two of the four SH3 domains of the E3 ubiquitin-protein ligase SH3RF1 (Uniprot Q7Z6J0; also known as 'plenty of SH3 domains' (POSH)), SH3-1 (135–194) and SH3-4 (829–888), were subcloned into a pESPRIT vector[42]. The constructs therefore contained an N-terminal hexahistidine tag followed by a tobacco etch virus (TEV) cleavage site. The final proteins after protease cleavage contain N-terminal GRR (POSH) or GHM extensions (JIP1).

To obtain unlabelled proteins, *Escherichia coli* BL21(DE3) cells transformed with one of the constructs were grown in lysogeny broth (LB) medium at 37 °C until the optical density at 600 nm reached 0.7. Protein expression was induced by the addition of isopropyl β-D-1-thiogalactopyranoside (IPTG) to a final concentration of 1 mM. The cultures were grown for an additional 15 h at 20 °C (POSH-SH3s) or 4 h at 37 °C (JIP1-SH3). The cells were collected by centrifugation and frozen at −20 °C or −80 °C. Isotopically $^{15}$N/$^{13}$C- and $^{15}$N-labelled proteins were produced by growing transformed *E. coli* BL21(DE3) cells in M9 minimal medium containing 1 g l$^{-1}$ of $^{15}$N-NH$_4$Cl and 2 g l$^{-1}$ of $^{13}$C$_6$-D-glucose or $^{12}$C$_6$-D-glucose. To obtain $^{15}$N-labelled protein with tyrosine residues site-selectively labelled at the ε position with $^{13}$C, transformed *E. coli* BL21(DE3) cells were grown in M9 minimal medium containing 1 g l$^{-1}$ of $^{15}$N-NH$_4$Cl, 2 g l$^{-1}$ of NaH$^{13}$CO$_3$ and 2 g l$^{-1}$ of [2-$^{13}$C]-glycerol[43].

All SH3 domains were purified by Ni affinity chromatography followed by size-exclusion chromatography. Cell lysis was carried out by sonication using purification buffer (POSH: 50 mM Tris pH 7.0/8.0, 500 mM NaCl, 1 mM β-mercaptoethanol; JIP1: 50 mM HEPES pH 7.0, 150 mM NaCl) supplemented with protease inhibitor tablets (Roche). The washing buffer used for Ni affinity chromatography was the same as the purification buffer with the addition of 20 mM imidazole. The elution buffer was the same as the purification buffer with the addition of 500 mM (POSH) or 300 mM (JIP1) imidazole. Nickel affinity chromatography was followed by cleavage by the TEV protease, a second Ni affinity column and size-exclusion chromatography on a Superdex 75 (GE Healthcare). This column was equilibrated with 50 mM HEPES pH 8.0, 500 mM NaCl, 2 mM DTT for POSH-SH3-1, 50 mM HEPES pH 7.0, 500 mM NaCl, 2 mM DTT for POSH-SH3-4 and 50 mM HEPES pH 7.0, 150 mM NaCl for JIP1-SH3.

## Expression and purification of JIP1-SH3 variants

Expression and purification of the JIP1-SH3 variants (Y526A, V517A, V517L, A541L and H493A) were performed following the same protocol as for JIP1-SH3, except that the cultures were grown for 15 h after induction at 20 °C (instead of 4 h at 37 °C).

## Thermal stability measurements of JIP1-SH3

The stability of JIP1-SH3 was measured by differential scanning fluorimetry using a Prometheus NT.48 (Nanotemper) instrument with the emission wavelengths set to 330 and 350 nm and an excitation power of 10%. The melting curve for wild-type JIP1-SH3 was measured at a protein concentration of 4 mg ml$^{-1}$ in 50 mM HEPES, 150 mM NaCl at pH 7.0 by using Prometheus Standard Capillaries (PR-C002). The temperature scan rate was fixed at 1 °C per min from 20 °C to 95 °C. The melting temperature ($T_m$) was calculated from the peak of the first derivative of the intrinsic protein fluorescence intensity ratio at 350 nm and 330 nm throughout the duration of the temperature ramp.

## NMR spectral assignment of JIP1-SH3 and its variants

The NMR assignment experiments were acquired in 50 mM HEPES, 150 mM NaCl, pH 7.0 at a protein concentration of 0.94 mM (JIP1-SH3), 1.06 mM (JIP1-SH3(Y526A)), 1.10 mM (JIP1-SH3(A541L)), 2 mM (JIP1-SH3(V517A)) and 0.90 mM (JIP1-SH3(H493A)). The NMR

spectral assignments of JIP1-SH3 were performed at 25 °C using a set of BEST-TROSY triple resonance experiments (HNCO, intra-residue HNCACO, HNCOCA, intra-residue HNCA, HNCOCACB and intra-residue HNCACB) acquired at a $^1$H frequency of 600 MHz (Bruker, operated with TopSpin v.3.5)[44]. The NMR spectral assignments of JIP1-SH3(Y526A) were obtained at 25 °C at a $^1$H frequency of 700 MHz (Bruker) using BEST-TROSY HNCO, HNCOCACB and intra-residue HNCACB experiments. The NMR spectral assignments of JIP1-SH3(A541L) (at 25 °C), JIP1-SH3(V517A) (at 35 °C) and JIP1-SH3(H493A) (at 35 °C) were obtained at a $^1$H frequency of 700 MHz (Bruker) using a BEST-TROSY HNCACB experiment. The spectra were manually peak-picked using NMRFAM-Sparky[45] and sequential connectivities were identified manually or by using the assignment program MARS[46]. Secondary structure propensities were calculated using SSP on the basis of the experimental Cα and Cβ chemical shifts[47].

## $^{15}$N relaxation measurements of JIP1-SH3

Measurements of $^{15}$N relaxation rates ($R_1$, $R_2$ and heteronuclear NOEs) of JIP1-SH3 were obtained using standard HSQC-type pulse sequences[48] at a $^1$H frequency of 600 MHz (Agilent, operated with VnmrJ v.3.1). The relaxation rates were measured at four different temperatures: 15, 25, 35 and 45 °C. The magnetization decay was sampled at (0, 100, 200, 400, 600, 800, 1,100, 1,500 and 1,900) milliseconds (ms) for longitudinal and at (10, 30, 50, 70, 90, 130, 170, 210 and 250) ms for transverse relaxation. Technical replicates of one or two of these delays were acquired to estimate the uncertainty on the relaxation rates using a Monte Carlo approach. Details of the Lipari–Szabo model free analysis can be found in the Supplementary Discussion.

## $^{15}$N and $^1$H$^N$ CPMG relaxation dispersion of JIP1-SH3 and its variants

All $^{15}$N CPMG relaxation dispersion experiments[24] were carried out at 15 °C using a constant-time relaxation delay of 32 ms with CPMG frequencies ($\nu_{CPMG}$) ranging from 31.25 to 1,000 Hz and a $^1$H decoupling field of 11 kHz. The $^1$H$^N$ relaxation dispersion experiments were carried out at 15 °C using the published pulse sequence[25] with a constant-time relaxation delay of 20 ms and CPMG frequencies ranging from 50 to 2,000 Hz. Uncertainties on peak intensities extracted from the relaxation dispersion experiments were estimated using the pooled s.d. calculated from repeat measurements (technical replicates of one to three $\nu_{CPMG}$ values), each pool being the set of repeat points per $\nu_{CPMG}$ and per peak. Uncertainties on $R_{2eff}$ values were propagated from the peak intensity uncertainty using a Monte Carlo approach. The following relaxation dispersion experiments were acquired: JIP1-SH3: $^{15}$N (600 MHz, Agilent), $^{15}$N (850 MHz, Bruker), $^1$H$^N$ (600 MHz, Bruker), $^1$H$^N$ (950 MHz, Bruker); JIP1-SH3(H493A) and JIP1-SH3(V517A): $^{15}$N (600 MHz, Bruker), $^{15}$N (950 MHz, Bruker), $^1$H$^N$ (600 MHz, Bruker), $^1$H$^N$ (950 MHz, Bruker); JIP1-SH3(Y526A): $^{15}$N (700 MHz, Bruker) and $^1$H$^N$ (600 MHz, Bruker); JIP1-SH3(A541L): $^{15}$N (700 MHz, Bruker). All relaxation dispersion data were analysed using the program ChemEx (https://github.com/gbouvignies/ChemEx)[49] as described in the Supplementary Discussion.

## Tyrosine assignments and $^{13}$C CPMG relaxation dispersion of Y526

The $^{13}$Cε–$^1$Hε tyrosine resonances were assigned at 45 °C by acquiring a two-dimensional (2D) plane of a BEST-TROSY intra-residue HNCACB experiment[44], an aromatic BEST constant-time $^1$H–$^{13}$C HSQC experiment[50] and a (Hβ)Cβ(CγCδCε)Hε experiment[51] linking the Cβ chemical shifts directly to the Hε chemical shifts. The spectra were manually peak-picked using NMRFAM-Sparky[45] and $^{13}$Cε–$^1$Hε tyrosine resonances were assigned manually (Supplementary Fig. 1). The acquisition of aromatic BEST constant-time $^1$H–$^{13}$C HSQC experiments at different temperatures (between 5 and 45 °C) enabled the final assignment at 15 °C.

Aromatic L-optimized TROSY-selected $^{13}$C CPMG[38] and $R_{1\rho}$[39] relaxation dispersion experiments were carried out at 15 °C on a 1 mM uniformly $^{15}$N and site-selective $^{13}$C-labelled JIP1-SH3 sample in 50 mM

HEPES, 150 mM NaCl at pH 7.0. CPMG relaxation dispersion experiments were carried out at magnetic field strengths of 600 MHz, 700 MHz and 850 MHz (Bruker) using a constant-time relaxation delay of 20 ms with CPMG frequencies ranging from 100 to 1,000 Hz. The $R_{1\rho}$ relaxation dispersion experiment was recorded on-resonance with Y526 at 700 MHz (Bruker) using $B_1$ field strengths ranging from 700 to 10,000 Hz with a 20 ms relaxation delay. Error bars were derived from repeat measurements as described above for the $^{15}$N and $^1$H$^N$ relaxation dispersion experiments. Analysis of the $^{13}$C data was carried out using ChemEx or using available analytical expressions for $R_{1\rho}$ relaxation in the presence of two-site exchange[52] (Supplementary Discussion).

## Comparison of JIP1-SH3 to other human SH3 domains

The sequences of 320 human SH3 domains were obtained[33], aligned using Clustal Omega[53] and categorized according to the identity of the amino acid at the position of Y526 in JIP1-SH3. A new alignment was performed using only the sequences that carry Y or F at this position, amounting to a total of 33 human SH3 domains. The sequences of the SH3 domains of the RIMS-binding proteins 1, 2 and 3 and the metastasis-associated in colon cancer protein 1 (MACC1) were not included in this alignment, as they contain longer insertions compared to the JIP1-SH3 sequence. For each SH3 domain, the amino acids corresponding to residues 493, 517 and 541 of JIP1-SH3 were assigned a size score according to the number of heavy atoms in their side chains (A: 1, C: 2, D: 4, E: 5, F: 7, G: 0, H: 6, I: 4, K: 5, L: 4, M: 4, N: 4, P: 3, Q: 5, R: 7, S: 2, T: 3, V: 3, Y: 8, W:10). A PCA was carried out to reveal potential correlations between the sizes of the amino acids in position 493, 517 and 541 using the ClustVis webtool[54].

## Crystallization of JIP1-SH3 and the variants Y526A, V517A, V517L, A541L and H493A

JIP1-SH3 and its variants were concentrated to a final concentration of 20 mg ml$^{-1}$ (JIP1-SH3, JIP1-SH3(V517A), JIP1-SH3(A541L) and JIP1-SH3(V517L)), 10 mg ml$^{-1}$ (JIP1-SH3(Y526A)) and 4 mg ml$^{-1}$ (JIP1-SH3(H493A)) after size-exclusion chromatography by using Amicon Ultra-4 3.0-kDa centrifugal filters (Merck). All crystals were obtained in 0.1 M HEPES pH 7.5, 1–5% PEG 400 and 2–2.5 M ammonium sulfate at 20 °C by the hanging-drop vapour diffusion method in 24-well plates (Hampton research)[55]. Drops of 2–3 µl consisting of 1:1 or 2:1 parts of protein solution and reservoir solution were vapour-equilibrated against 500 µl of reservoir solution. All crystals appeared after two days and were collected by transferring them to a mother liquor solution containing 20–30% trehalose, frozen and kept in liquid nitrogen.

## Crystallization of POSH-SH3-1 and POSH-SH3-4

Purified POSH-SH3-1 and POSH-SH3-4 were directly concentrated after size-exclusion chromatography to 3.4 and 5.0 mg ml$^{-1}$, respectively, using Amicon Ultra-4 3.5-kDa centrifugal filters (Merck). Initial crystallization conditions were identified using the high-throughput crystallization platform (EMBL).

The initial condition identified for POSH-SH3-1 was 0.2 M NaF, 20% PEG 3350 from the PEGs-I screen (Qiagen) at 4 °C. Needles appeared after 3 to 7 days. Further optimization was done using the hanging-drop vapour diffusion method at 4 °C in 24-well plates. Drops of 2 µl consisting of equal parts protein solution at 2.5 mg ml$^{-1}$ and reservoir solution (0.2 M NaF, 22% PEG 3350) were vapour-equilibrated against 500 µl of reservoir solution. Hexagonal crystals appeared after three days and were collected after five days by transferring them to a mother liquor solution containing 5% ethylene glycol as cryoprotectant, frozen and kept in liquid nitrogen.

The initial screen of POSH-SH3-4 identified two crystallization conditions: 0.1 M MES pH 6.5, 25% PEG 3000 (condition 1) and 0.1 M MES pH 6.5, 25% PEG 4000 (condition 2) from the PEGs-I screen (Qiagen)

at 4 °C. Diffraction-quality needles (condition 1, 0.1 M MES pH 6.5, 26% PEG 3000) or three-dimensional crystals (condition 2, 0.1 M MES pH 6.5, 23% PEG 4000) were obtained after four days using the same vapour-diffusion set-up as for POSH-SH3-1. These were collected after seven days with 10% ethylene glycol as cryoprotectant, frozen and kept in liquid nitrogen.

## Structure determination

Crystal diffraction was performed at the ESRF beamlines ID30A, ID23-1, ID23-2 using the MXCube software[56,57], at the automated beamline MASSIF-1[58] or at the Diamond beamlines I04 and I04-1, all equipped with Pilatus detectors (Dectris). Indexing and integration was performed using the XDS[59], the autoProc[60] or GrenADeS[61] program suites. Data reduction for JIP1-SH3(H493A) was carried out with Pointless and Aimless[62,63]. Molecular replacement of the wild-type JIP1-SH3 structure was carried out in Phaser[64] using the PDB code 2FPE (chains A–B) as a search model. The structures of JIP1-SH3 mutants were obtained by using our wild-type JIP1-SH3 structure as a search model. The initial solutions were improved through cycles of manual adjusting in Coot[65] and refined by using Refmac5[66]. Aimless, Phaser and Refmac were all used as programs of the CCP4 suite[67].

The structure of POSH-SH3-4 was determined by molecular replacement using a homology model that was built on the basis of the SH3 domain structure of SORBS1 (PDB code: 2LJ1, chain A), which has 45% sequence identity. The structure of POSH-SH3-1 was determined by molecular replacement using as a search model the SH3 domain of human tyrosine protein kinase C-Src (PDB code: 2SRC). Crystallography applications were compiled and configured by SBGrid[68].

## Structural trajectory and void volume calculations

The structural trajectory between the major and minor conformation was generated with Chimera[69] by morphing between the wild-type JIP1-SH3 structure (PDB 7NYK) and the structures of the two variants JIP1-SH3(H493A) (7NYL) and JIP1-SH3(V517A) (7NYM). To calculate changes in the volume of the Y526 pocket, protons were added to all structures of the trajectory and Y526 was replaced by glycine to allow calculation of the complete pocket volume by POVME 3.0 using a distance cut-off of 1.09 Å, corresponding to the van der Waals radius of a hydrogen atom[70]. A similar strategy was used to generate the structural trajectory between the wild-type JIP1-SH3 structure (PDB 7NYK) and the structure of the JIP1-SH3(A541L) variant (7NYO).

## MD simulations of JIP1-SH3

MD simulations were carried out using ACEMD v.3.3.0[71] and the Charmm36m force field parameters[72]. Using VMD[73], coordinates of the dimer from PDB 2FPE were inserted in the box of dimensions with a minimum distance of 2 Å in each direction between each atom and any box side. The box was then filled with water molecules and an amount of Na$^+$ and Cl$^-$ corresponding to [NaCl] = 0.1 M. Electrostatic interactions were evaluated using Particle-Mesh Ewald (PME) electrostatics with a cut-off distance of 9 Å. Van der Waals forces were calculated with a cut-off of 9 Å and a switching function active from 7.5 Å to smoothly reduce the potential to zero. An integration step of 2 fs and holonomic constraints on all hydrogen-heavy atom bond terms were used. The energy of the system was minimized using conjugate-gradient minimization for 500 steps. Random velocities from a Maxwell distribution with $T$ = 298.15 K were assigned to atoms. Then, the system was equilibrated first for 100 ps in the NVE ensemble and then for 1 ns in the NPT ensemble. In the latter case, temperature and pressure were controlled using the Langevin thermostat with a damping constant of 1 ps$^{-1}$ and Berendsen barostat with a relaxation time of 400 fs, respectively. Finally, a 1 µs trajectory was calculated in the NVT ensemble using the Langevin thermostat with a damping constant of 0.1 ps$^{-1}$. Trajectories were processed and analysed using the MDAnalysis Python package[74].

## Reporting summary

Further information on research design is available in the Nature Research Reporting Summary linked to this paper.

## Data availability

Protein structure data have been deposited in the PDB with accession codes: 7NYK (JIP1-SH3), 7NZB (JIP1-SH3(V517L)), 7NYO (JIP1-SH3(A541L)), 7NYL (JIP1-SH3(H493A)), 7NYM (JIP1-SH3(V517A)), 7NZC (POSH-SH3-1) and 7NZD (POSH-SH3-4). The $^1$H, $^{13}$C and $^{15}$N chemical shifts of JIP1-SH3 have been deposited in the Biological Magnetic Resonance Data Bank with accession codes: 50814 (JIP1-SH3), 50817 (JIP1-SH3(Y526A)), 50816 (JIP1-SH3(V517A)), 50818 (JIP1-SH3(H493A)) and 50815 (JIP1-SH3(A541L)). SH3 domain structures for molecular replacement were retrieved from the PDB (https://www.ebi.ac.uk/pdbe/) with accession codes: 2FPE (JIP1), 2LJ1 (SORBS1) and 2SRC (tyrosine protein kinase C-Src). Structures for the proteome-wide SH3 sequence analysis were retrieved from the PDB with accession codes: 1CSK, 3A98, 2O9S, 5VEI and 4LNP.

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

**Acknowledgements** We thank W. Adamski, S. Milles and T. Herrmann for discussions. We acknowledge the platforms of the Grenoble Instruct European Research Infrastructure Consortium (Integrated Structural Biology Grenoble; UAR 3518 CNRS-CEA-UGA-EMBL) within the Grenoble Partnership for Structural Biology. Platform access was supported by French Infrastructure for Integrated Structural Biology (ANR-10-INBS-05-02) and the Grenoble Alliance for Integrated Structural and Cell Biology, a project of the University Grenoble Alpes graduate school (Ecoles Universitaires de Recherche) CBH-EUR-GS (ANR-17-EURE-0003). The Institut de Biologie Structurale acknowledges integration into the Interdisciplinary Research Institute of Grenoble. The authors thank the Diamond Light Source and the European Synchrotron Radiation Facility for beamtime access and technical support. This work was funded by the Fondation pour la Recherche Medicale through a postdoctoral fellowship to L.M.P. (contract SPF201909009258) and by a PhD fellowship to J.K. from the IDPbyNMR Marie Curie action of the European commission (contract no. 264257). Funding is also acknowledged from the Agence National de la Recherche (ANR) through ANR T-ERC MAPKassembly (to M.R.J.), ANR ScaffoldDisorder (to M.R.J. and A.P.) and ANR JCJC RC18114CC NovoTargetParasite (to A.P.).

**Author contributions** L.M.P., M.R.J. and A.P. conceived the study. L.M.P., L.M.B., D.M. and J.K. made samples. L.M.B. and A.P. solved the structures of the POSH SH3 domains. L.M.P., F.S.I. and A.P. solved the structures of JIP1-SH3 and its variants. L.M.P., J.K. and M.R.J. designed and performed all NMR experiments. L.M.P., G.B. and M.R.J. analysed and interpreted the NMR data. M.B. and N.S. performed and analysed the MD simulation. M.R.J. and A.P. wrote the paper with input from all authors.

**Competing interests** The authors declare no competing interests.

**Additional information**
**Correspondence and requests for materials** should be addressed to Andrés Palencia or Malene Ringkjøbing Jensen.

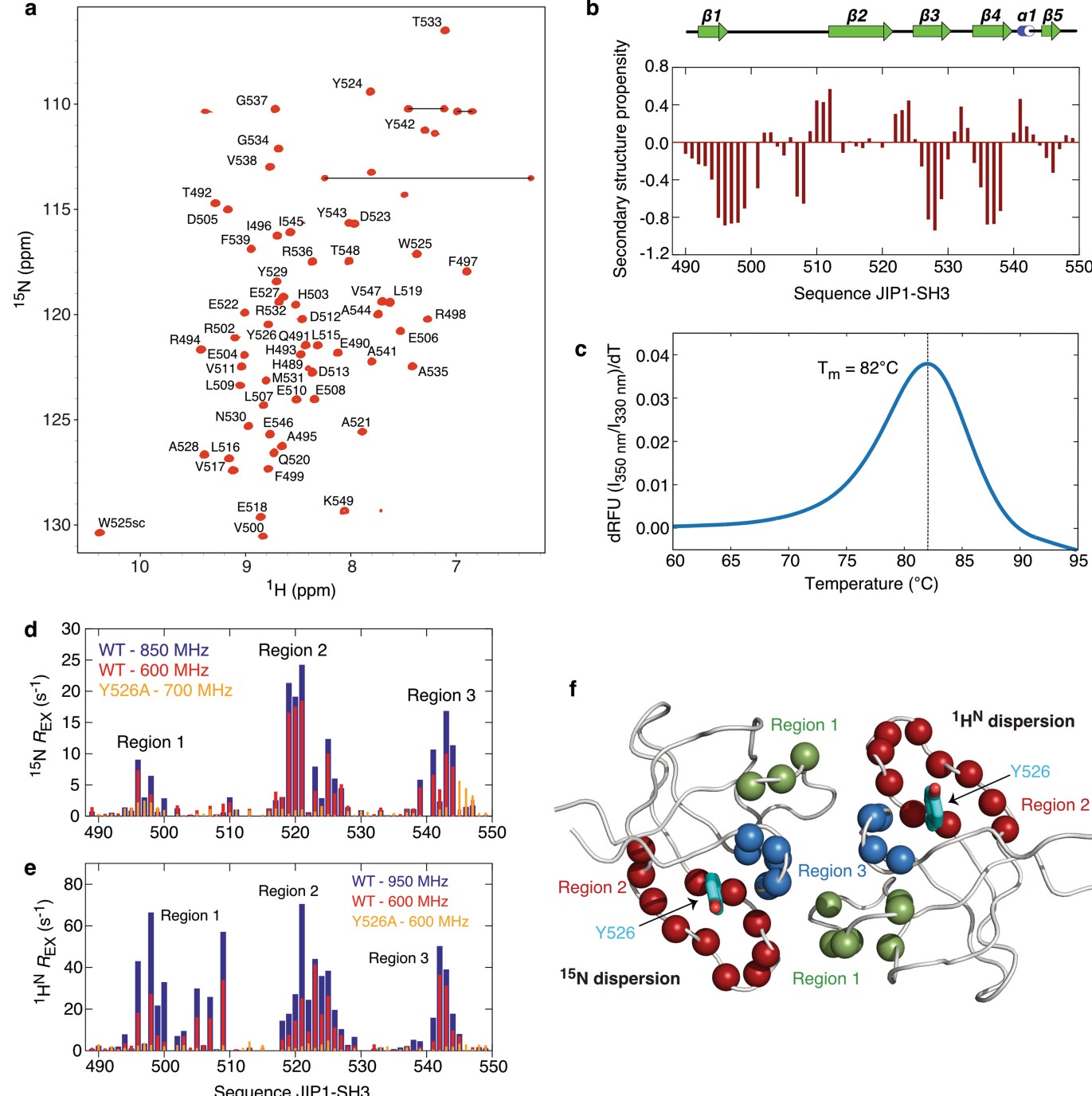

**Extended Data Fig. 1 | Structural propensities and conformational exchange in JIP1-SH3. a**, $^1H$–$^{15}N$ HSQC spectrum of JIP1-SH3 at 25 °C with labels indicating assignments. Horizontal lines connect side chain resonances. **b**, Secondary structure propensities of JIP1-SH3 calculated from experimental $^{13}C\alpha$ and $^{13}C\beta$ chemical shifts at 25 °C. The position of secondary structure elements is indicated, as observed in the crystal structure of JIP1-SH3 (PDB: 2FPE). **c**, Differential scanning fluorimetry (DSF) melting curve for JIP1-SH3 (RFU – relative fluorescence unit). **d**, Conformational exchange contributions, $R_{EX}$, extracted from $^{15}N$ CPMG relaxation dispersion data acquired at 15 °C of JIP1-SH3 (blue – 850 MHz, red – 600 MHz) and JIP1-SH3(Y526A) (orange – 700 MHz)

as the difference between $R_{2eff}$ at low (31 Hz) and high (1 kHz) CPMG frequencies. **e**, Exchange contributions, $R_{EX}$, extracted from $^1H^N$ CPMG relaxation dispersion data acquired at 15 °C of JIP1-SH3 (blue – 950 MHz, red – 600 MHz) and JIP1-SH3(Y526A) (orange – 600 MHz) as the difference between $R_{2eff}$ at low (50 Hz) and high (2 kHz) CPMG frequencies. **f**, Structure of the dimeric SH3 domain of JIP1 with Y526 shown in cyan. Residues with exchange contributions ($^{15}N R_{EX} > 3 s^{-1}$ and $^1H^N R_{EX} > 10 s^{-1}$) are shown as spheres with colours indicating their position in the primary sequence (region 1 – green, region 2 – red, region 3 – blue). The residues showing $^{15}N$ and $^1H^N$ exchange contributions are represented separately on each monomer.

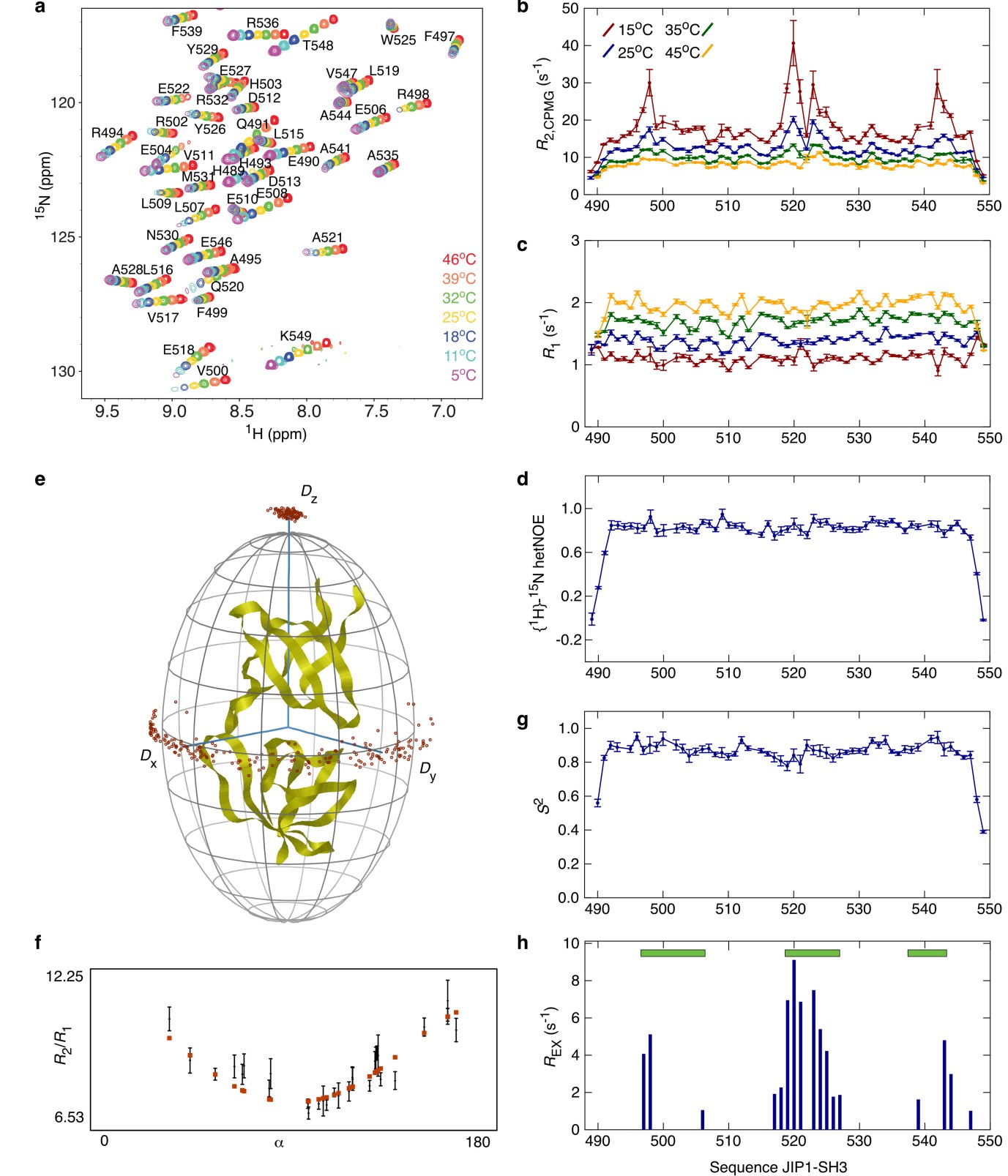

**Extended Data Fig. 2** | See next page for caption.

**Extended Data Fig. 2 | Lipari–Szabo model-free analysis of $^{15}$N relaxation data of JIP1-SH3. a**, Region of the $^1$H–$^{15}$N HSQC spectrum of JIP1-SH3 at temperatures ranging from 5 to 46 °C. **b**, Experimental $^{15}$N $R_2$ (CPMG) relaxation rates at a $^1$H frequency of 600 MHz and four different temperatures. **c**, Experimental $^{15}$N $R_1$ relaxation rates at a $^1$H frequency of 600 MHz and four different temperatures. **d**, Experimental {$^1$H}-$^{15}$N heteronuclear NOEs acquired at a $^1$H frequency of 600 MHz and 25 °C. Error bars in **b**–**d** represent one standard deviation (s.d.) derived from Monte Carlo simulations of experimental uncertainty. **e**, A model-free analysis of $^{15}$N $R_1$, $R_2$ and heteronuclear NOEs at 25 °C was carried out providing an axially symmetric diffusion tensor (Supplementary Discussion). The diffusion tensor is represented relative to the dimeric structure of JIP1-SH3. Distributions of axis orientations are shown as red dots and were determined from Monte Carlo

simulations using Tensor2[75]. **f**, Angular dependence of the $R_2/R_1$ ratios relative to the main axis of the diffusion tensor of JIP1-SH3. Only residues without exchange contributions to the transverse relaxation and for which the {$^1$H}-$^{15}$N NOE is above 0.7 were included in the analysis. Error bars are centred at experimental values and were propagated from the experimental uncertainty on $R_2$ and $R_1$. Orange squares are back-calculated values using the optimal tensor. **g**, Order parameters, $S^2$, derived from the model-free analysis of the relaxation data at 25 °C. Error bars represent one standard deviation (s.d.) derived from Monte Carlo simulations as implemented in Tensor2. **h**, Conformational exchange contributions, $R_{EX}$, derived from the model-free analysis of the relaxation data at 25 °C. Green bars indicate residues that are located in the dimer interface of the SH3 domain as detected by the PISA server[76].

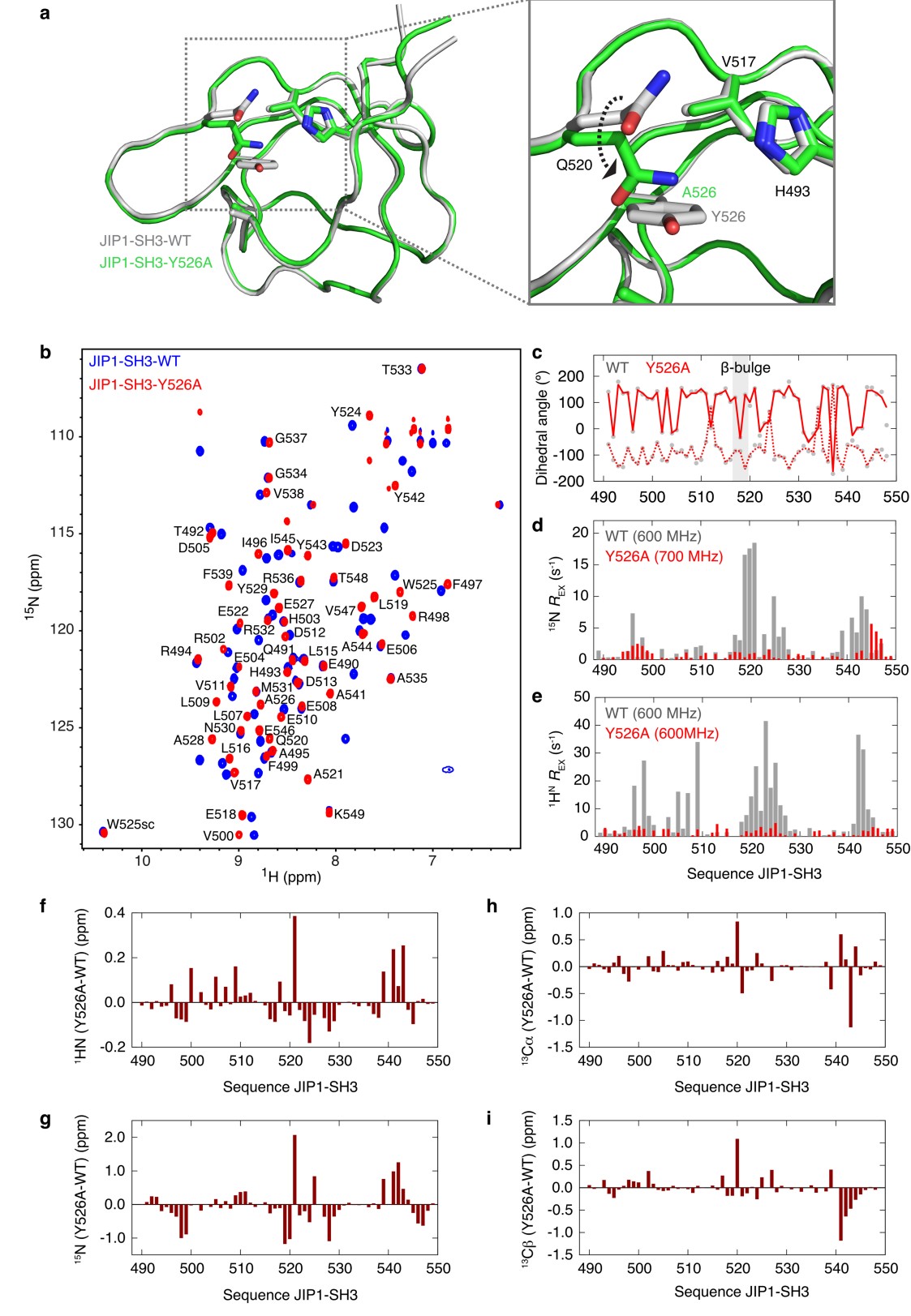

**Extended Data Fig. 3** | See next page for caption.

**Extended Data Fig. 3 | Structure and dynamics of the Y526A variant of JIP1-SH3.** **a**, Comparison of the crystal structure of JIP1-SH3(Y526A) (green) with the WT structure (grey) showing an almost identical backbone conformation of the two proteins. The zoom highlights a minor structural difference at the level of the side chain of Q520 which reorients in the variant to take up the position normally occupied by Y526 in the WT protein. **b**, Superposition of $^1H-^{15}N$ HSQC spectra of JIP1-SH3(Y526A) (red) and JIP1-SH3(WT) (blue) acquired at 25 °C. **c**, Comparison of dihedral angles in JIP1-SH3(WT) (grey spheres) and JIP1-SH3(Y526A) (red lines). Dashed lines correspond to the backbone φ angle and full drawn lines to the backbone ψ angle. **d**, Conformational exchange contributions, $R_{EX}$, extracted from $^{15}N$ CPMG relaxation dispersion data as the difference between $R_{2eff}$ at low (31 Hz) and high (1 kHz) CPMG frequencies. The exchange contributions are compared for WT JIP1-SH3 (grey, at 600 MHz) and the Y526A variant (red, at 700 MHz). **e**, Conformational exchange contributions, $R_{EX}$, extracted from $^1H^N$ CPMG relaxation dispersion data as the difference between $R_{2eff}$ at low (50 Hz) and high (2 kHz) CPMG frequencies. The exchange contributions are compared for the WT JIP1-SH3 (grey, at 600 MHz) and the Y526A variant (red, at 600 MHz). **f–i**, Chemical shift differences between the Y526A variant and WT JIP1-SH3 for $^1H^N$ (**f**), $^{15}N$ (**g**), $^{13}C\alpha$ (**h**) and $^{13}C\beta$ (**i**).

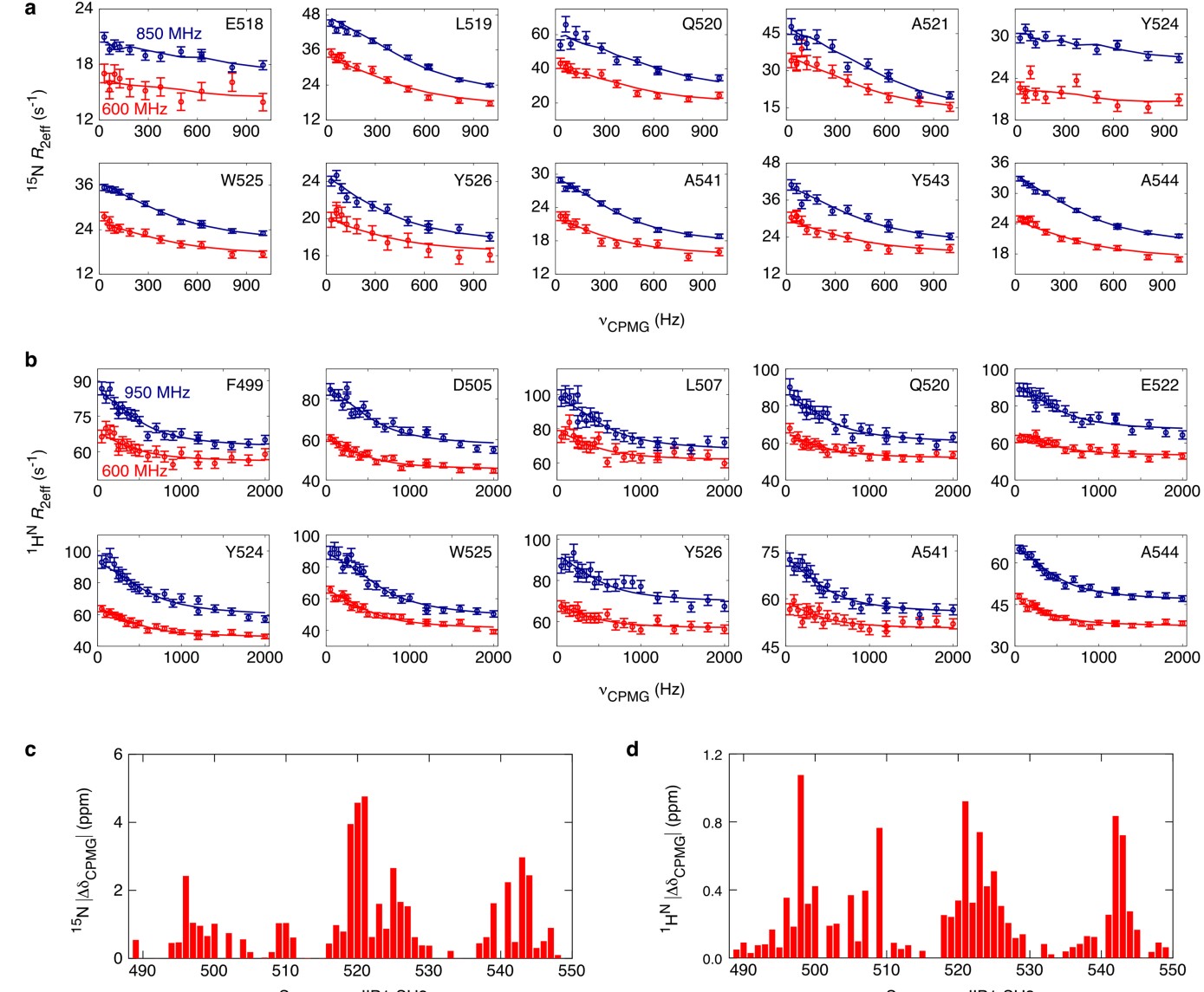

**Extended Data Fig. 4 | Analysis of $^{15}$N and $^1$H$^N$ relaxation dispersion data of JIP1-SH3. a**, Examples of $^{15}$N CPMG relaxation dispersion profiles for JIP1-SH3 obtained at two magnetic field strengths (red – 600 MHz, blue – 850 MHz) at 15 °C. **b**, Examples of $^1$H$^N$ CPMG relaxation dispersion profiles for JIP1-SH3 obtained at two magnetic field strengths (red – 600 MHz, blue – 950 MHz) at 15 °C. The $^{15}$N and $^1$H$^N$ data were analysed simultaneously for all residues according to a two-site exchange model (full-drawn lines in **a** and **b**). Error bars in **a** and **b** represent one standard deviation (s.d.) derived from Monte Carlo simulations of experimental uncertainty. **c**, **d**, Chemical shift differences between the major and minor state extracted from a simultaneous analysis of the relaxation dispersion data at 15 °C for $^{15}$N (**c**) and $^1$H$^N$ (**d**).

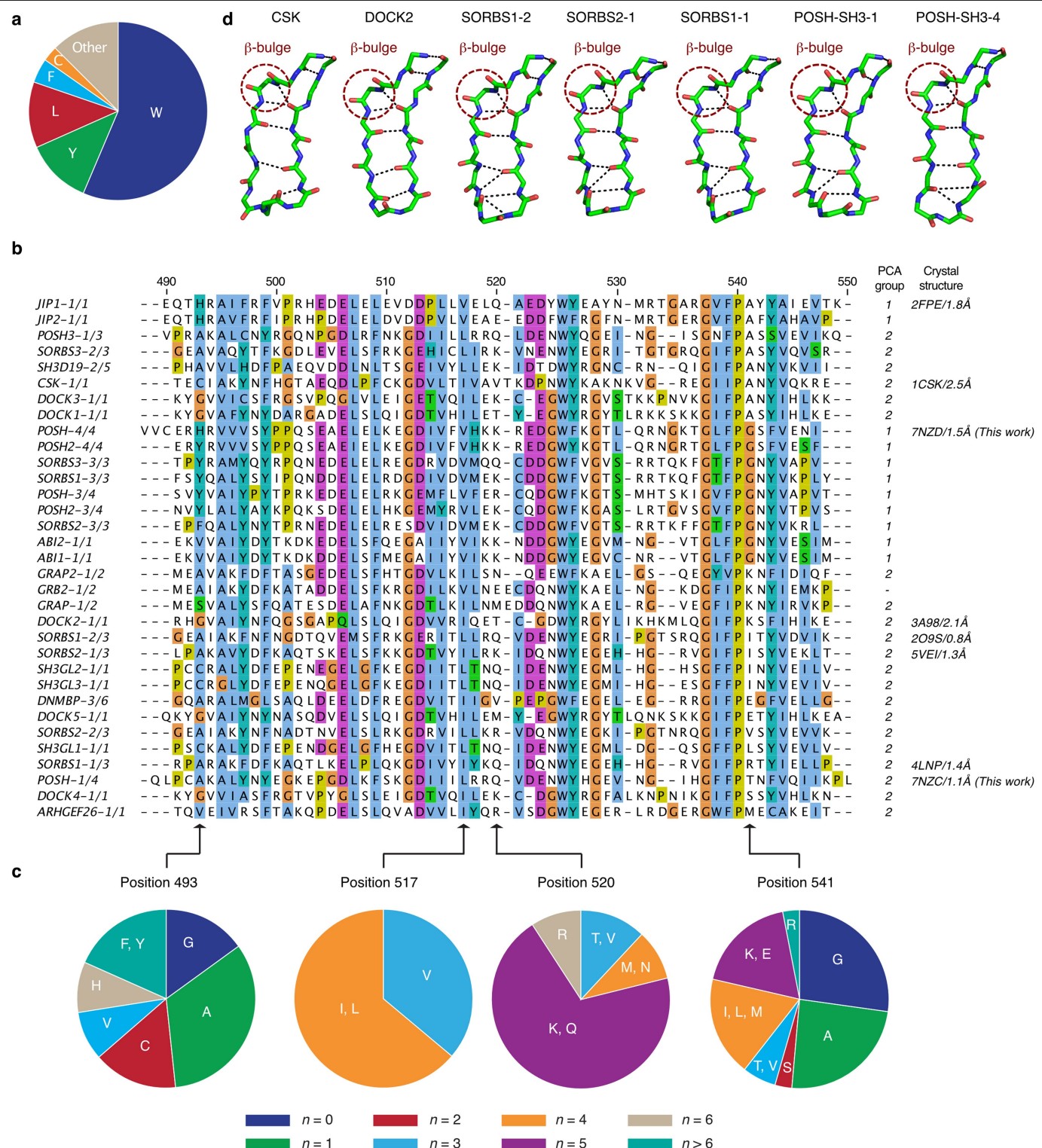

**Extended Data Fig. 5 | Analysis of the sequence composition of 320 human SH3 domains. a**, Distribution of amino acid types at the position of Y526 of JIP1-SH3 following a sequence alignment of all 320 SH3 domains. **b**, Sequence alignment of all identified SH3 domains carrying either a phenylalanine (F) or tyrosine (Y) at the position of Y526 in JIP1-SH3. For each sequence the PCA group is indicated along with the PDB code and resolution of available crystal structures. **c**, Pie charts showing the distribution of amino acid types in the different SH3 domains at positions corresponding to residue 493, 517, 520 and 541 in JIP1-SH3. The pie charts are colour-coded according to the size score assigned to each amino acid type corresponding to the number of heavy atoms in their side chains (see online Methods). **d**, Illustration of the backbone conformation of the β-sheet formed between the 516–521 and 524–529 regions in SH3 domains with tyrosine or phenylalanine at position 526. The β-bulge conformation observed in WT JIP1-SH3 is observed in all SH3 domains for which high-resolution crystal structures are available.

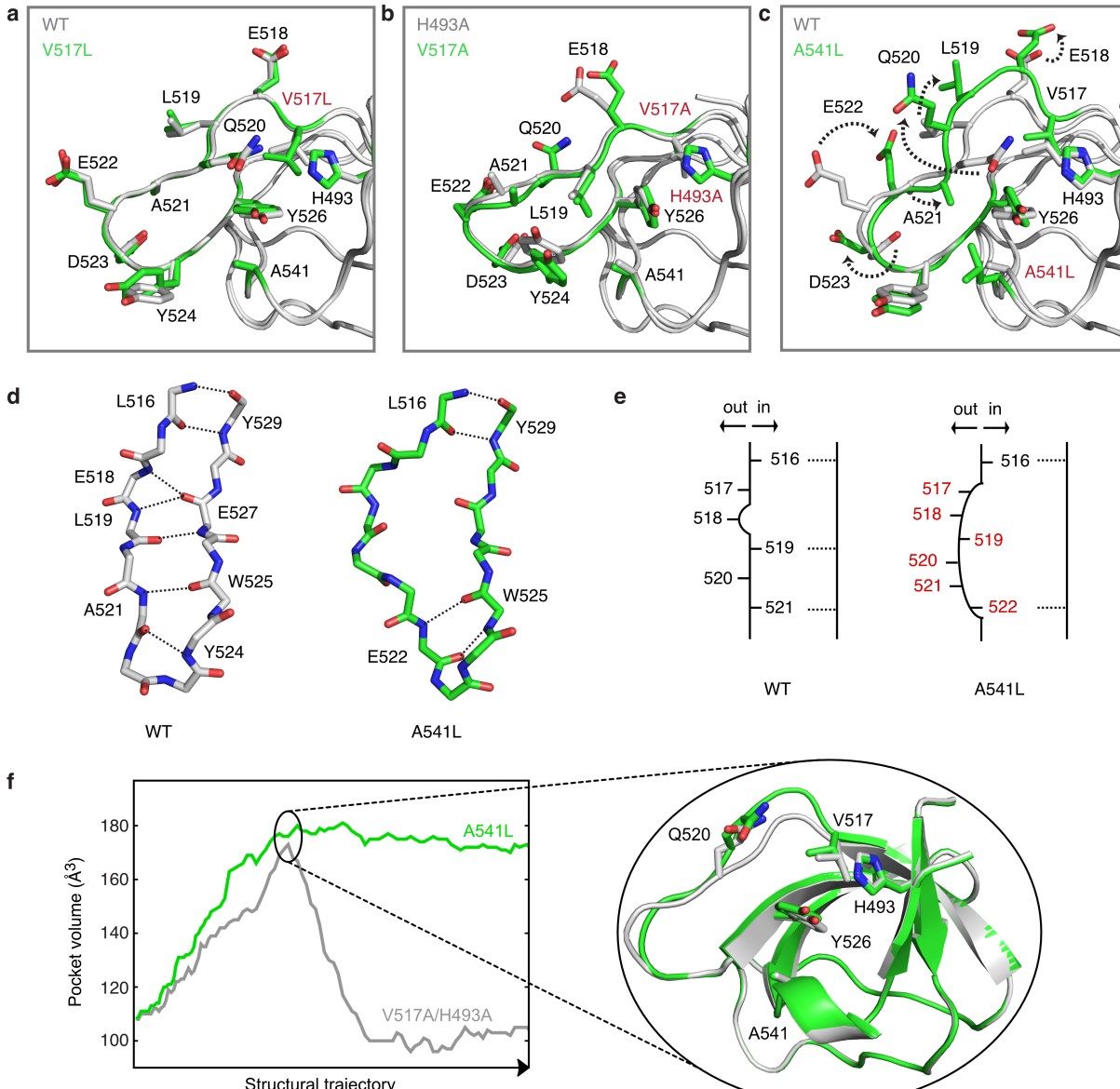

**Extended Data Fig. 6 | Comparison of the structures of JIP1-SH3 and different variants. a**, Comparison of the crystal structures of JIP1-SH3(WT) (grey) and its V517L variant (green). The backbone conformation is entirely conserved with only minor rearrangements of protein side chains. **b**, Comparison of the crystal structures of the H493A (grey) and V517A (green) variants of JIP1-SH3. Structural features are conserved including similar side chain conformations. **c**, Comparison of the crystal structures of JIP1-SH3(WT) (grey) and its A541L variant (green) with arrows indicating the major conformational rearrangements between the WT protein and the variant. **d**, Illustration of the backbone conformation of the β-sheet formed between the 516–521 and 524–529 regions in the WT protein (left), and in the A541L variant (right). Dashed lines indicate hydrogen bonds. The A541L variant does not

adopt a β-sheet conformation owing to the absence of several hydrogen bonds between the two β-strands. **e**, Schematic representation of the conformation of the β-strand encompassing residues 516 to 521 showing the orientation of the carbonyl group ("out" – carbonyl group surface exposed, "in" – carbonyl group pointing towards the β-strand encompassing residues 524 to 529) in the WT protein (left) and in the A541L variant (right). **f**, Volume of the Y526 pocket along the structural trajectory between the WT conformation and the conformations of the H493A/V517A mutants (grey) or A541L mutant (green). Conformations from the two structural trajectories at the maximum pocket volume are superimposed and shown in cartoon representation. The initial pocket expansion and its associated structural features are shared among all mutants.

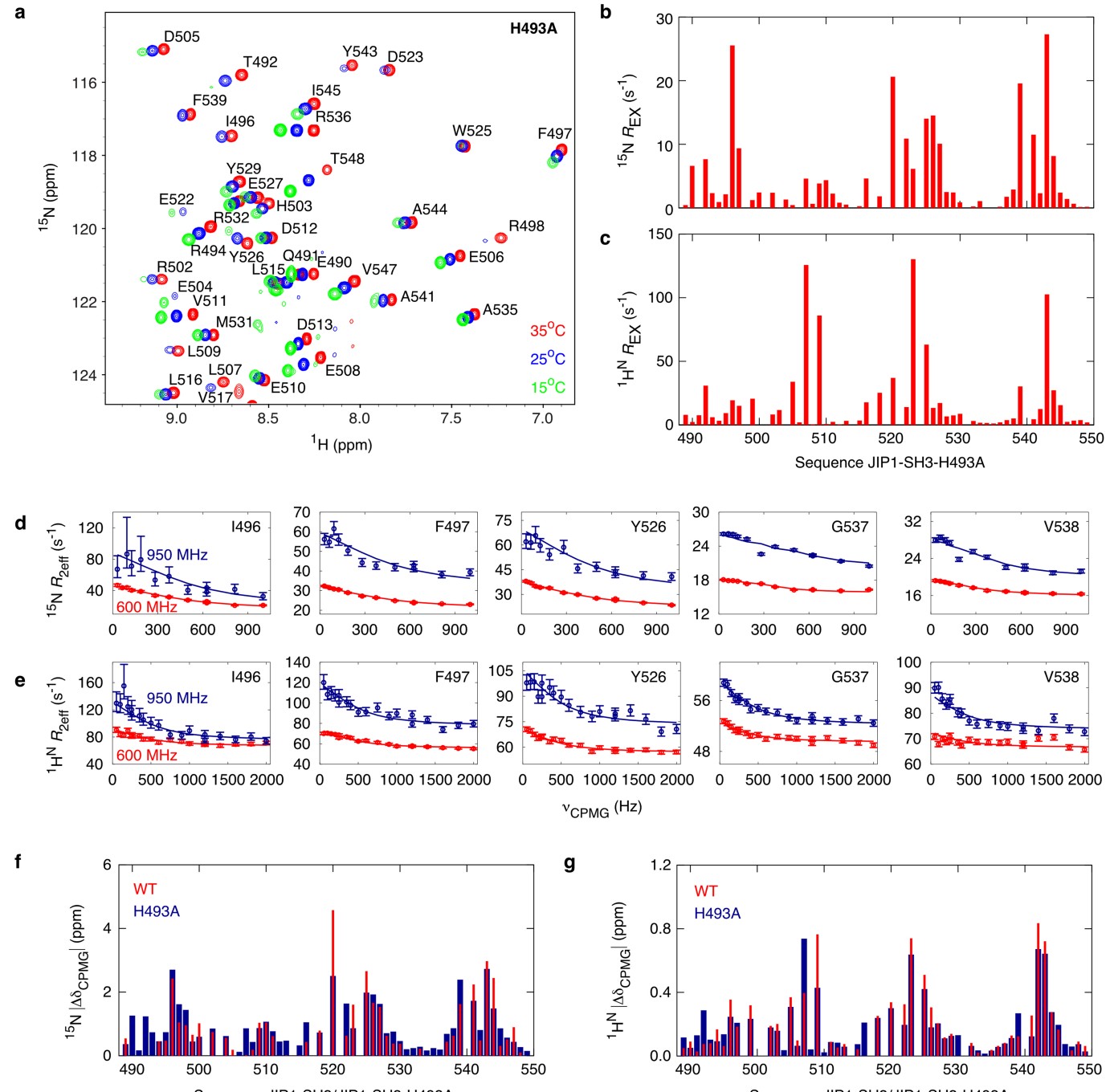

**Extended Data Fig. 7 | Analysis of CPMG relaxation dispersion data of the H493A variant of JIP1-SH3. a**, Region of the $^1$H-$^{15}$N HSQC of the H493A variant at three different temperatures (green - 15 °C, blue - 25 °C and red - 35 °C). **b**, Conformational exchange contributions, $R_{EX}$, extracted from $^{15}$N CPMG relaxation dispersion data of the H493A variant as the difference between $R_{2eff}$ at low (31 Hz) and high (1 kHz) CPMG frequencies (600 MHz and 15 °C). **c**, Conformational exchange contributions, $R_{EX}$, extracted from $^1$H$^N$ CPMG relaxation dispersion data of the H493A variant as the difference between $R_{2eff}$ at low (50 Hz) and high (2 kHz) CPMG frequencies (600 MHz and 15 °C). **d, e**, Examples of $^{15}$N (**d**) and $^1$H$^N$ (**e**) CPMG relaxation dispersion profiles of the

H493A variant of JIP1-SH3 obtained at two magnetic field strengths (red – 600 MHz, blue – 950 MHz) at 15 °C. The $^{15}$N and $^1$H$^N$ data were analysed simultaneously for all residues according to a two-site exchange model (full-drawn lines in **d** and **e**) using a population of the minor state fixed to 10%. Error bars represent one standard deviation (s.d.) derived from Monte Carlo simulations of experimental uncertainty. **f, g**, Comparison of the chemical shift differences between the major and minor state extracted from relaxation dispersion experiments for WT JIP1-SH3 (red) and its H493A variant (blue). Data are shown for both $^{15}$N (**f**) and $^1$H$^N$ (**g**) chemical shifts.

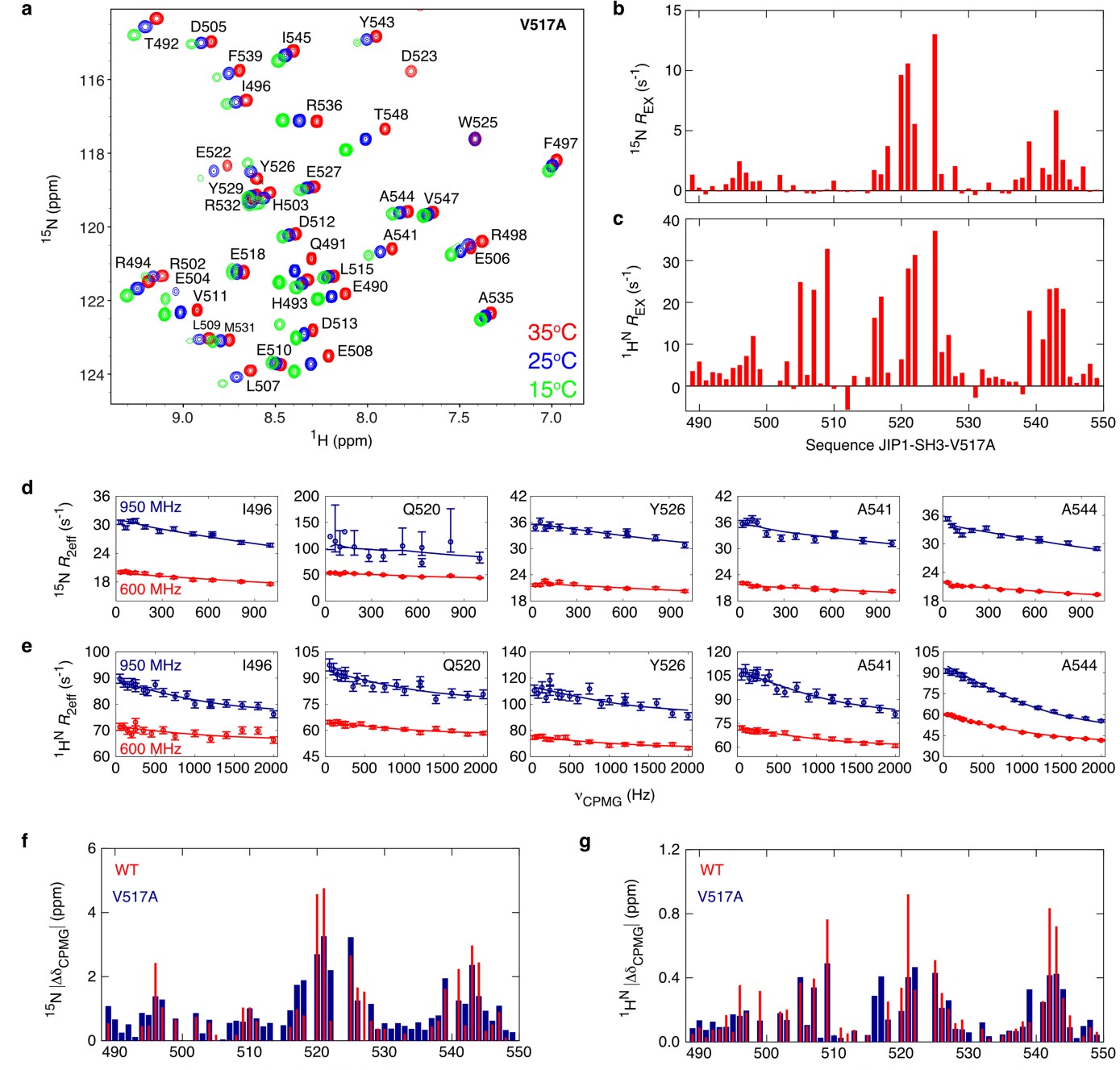

**Extended Data Fig. 8 | Analysis of CPMG relaxation dispersion data of the V517A variant of JIP1-SH3. a**, Region of the $^1$H–$^{15}$N HSQC of the V517A variant at three different temperatures (green - 15 °C, blue - 25 °C and red - 35 °C). **b**, Conformational exchange contributions, $R_{EX}$, extracted from $^{15}$N CPMG relaxation dispersion data of the V517A variant as the difference between $R_{2eff}$ at low (31 Hz) and high (1 kHz) CPMG frequencies (600 MHz and 15 °C). **c**, Conformational exchange contributions, $R_{EX}$, extracted from $^1$H$^N$ CPMG relaxation dispersion data of the V517A variant as the difference between $R_{2eff}$ at low (50 Hz) and high (2 kHz) CPMG frequencies (600 MHz and 15 °C). **d**, **e**, Examples of $^{15}$N (**d**) and $^1$H$^N$ (**e**) CPMG relaxation dispersion profiles of the V517A

variant of JIP1-SH3 obtained at two magnetic field strengths (red – 600 MHz, blue – 950 MHz) at 15 °C. The $^{15}$N and $^1$H$^N$ data were analysed simultaneously for all residues according to a two-site exchange model (full-drawn lines in **d** and **e**) using a population of the minor state fixed to 11%. Error bars represent one standard deviation (s.d.) derived from Monte Carlo simulations of experimental uncertainty. **f**, **g**, Comparison of the chemical shift differences between the major and minor state extracted from relaxation dispersion experiments for WT JIP1-SH3 (red) and its V517A variant (blue). Data are shown for both $^{15}$N (**f**) and $^1$H$^N$ (**g**) chemical shifts.

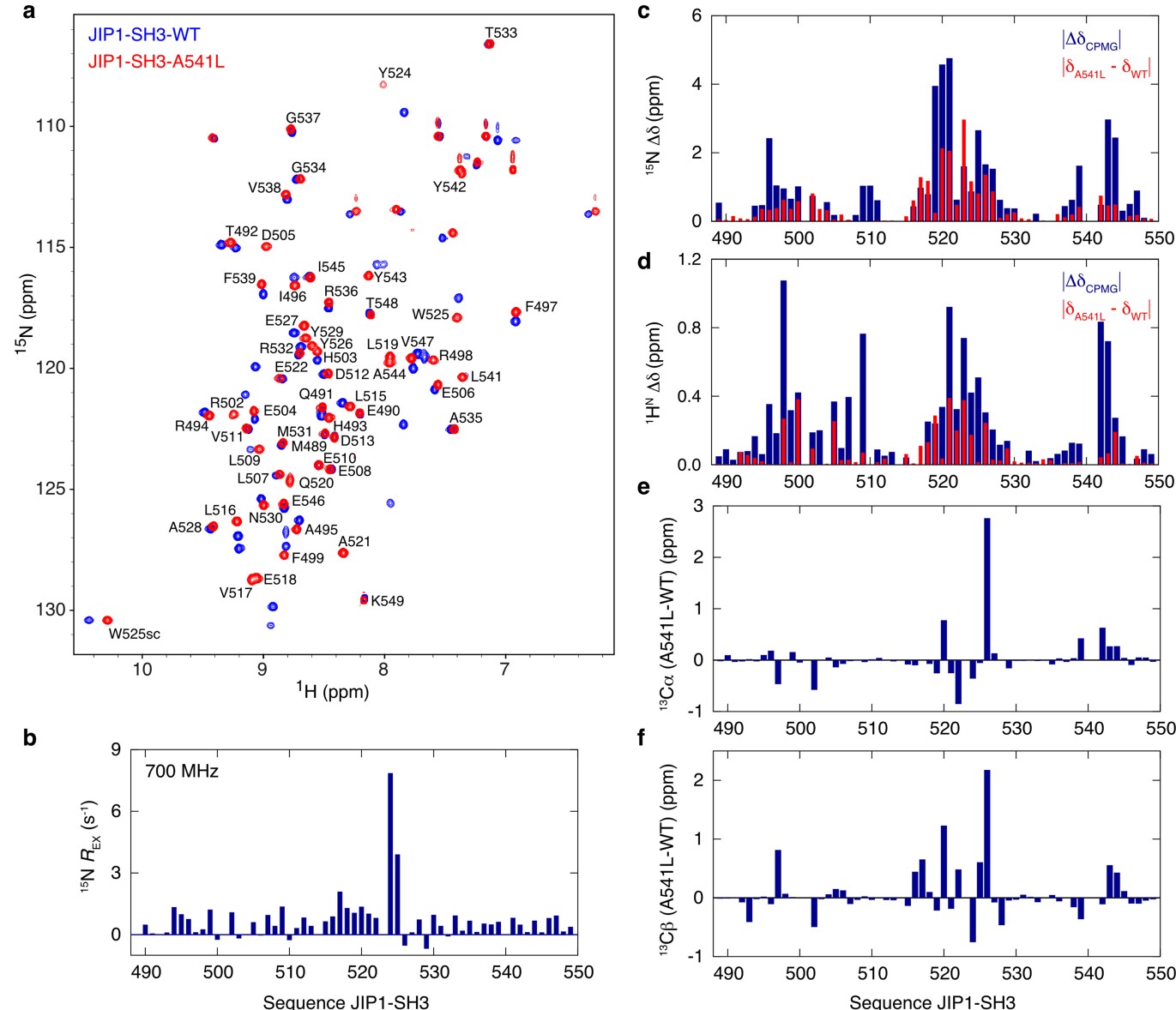

**Extended Data Fig. 9 | The A541L variant does not capture the structural details of the minor state detected by CPMG relaxation dispersion in JIP1-SH3. a**, Superposition of $^1$H–$^{15}$N HSQC spectra of JIP1-SH3(A541L) (red) and WT JIP1-SH3 (blue) acquired at 15 °C. **b**, Conformational exchange contributions, $R_{2eff}$, extracted from $^{15}$N CPMG relaxation dispersion data of the A541L variant as the difference between $R_{2eff}$ at low (31 Hz) and high (1 kHz) CPMG frequencies at 700 MHz and 15 °C. Only modest $^{15}$N conformational exchange contributions are observed suggesting that this variant is populating a single conformation in solution represented by the determined crystal structure. **c**, **d**, Comparison of the chemical shift differences between the major and minor states extracted from relaxation dispersion experiments of WT JIP1-SH3 (blue) and the chemical shift differences between the observed chemical shifts of the A541L variant and WT JIP1-SH3 (red). Data are shown for $^{15}$N (**c**) and for $^1$H$^N$ (**d**) nuclei. The poor agreement between the two datasets show that the A541L crystal structure is not representative of the conformation of the minor state detected by NMR relaxation dispersion. **e**, **f**, Chemical shift differences between the A541L variant and WT JIP1-SH3 for $^{13}$Cα (**e**) and $^{13}$Cβ (**f**).

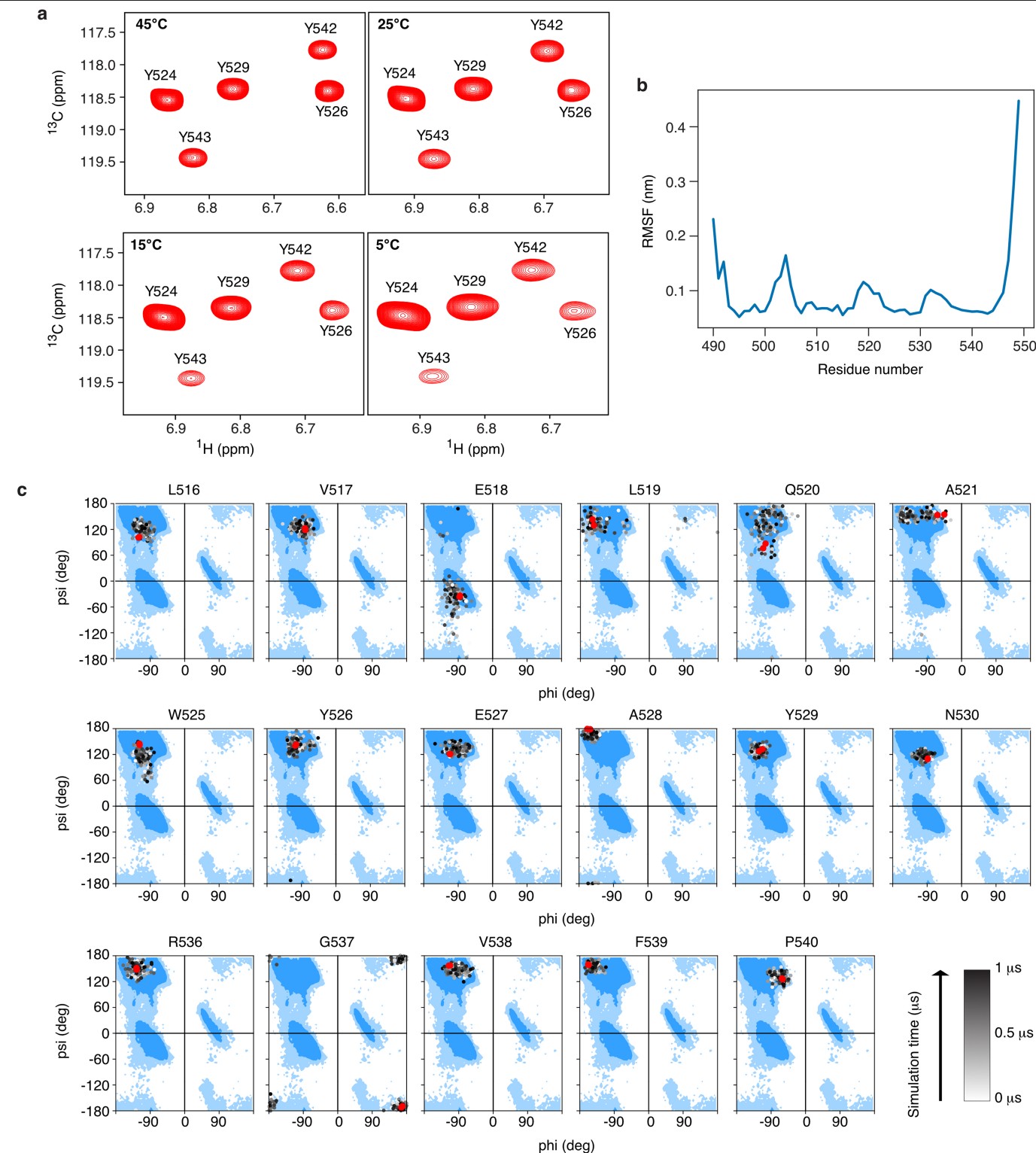

**Extended Data Fig. 10 | The aromatic ring of Y526 undergoes fast flipping.**
**a**, ¹H–¹³C HSQC spectra of JIP1-SH3 showing the region of tyrosine epsilon
correlations. A sample of JIP1-SH3 site-selectively labelled at the epsilon
positions was used, and spectra at four different temperatures were acquired
in the range from 5 to 45 °C. **b**, Per-residue root-mean square fluctuation
(RMSF) of Cα atoms during the MD simulation. **c**, Residue-specific
Ramachandran plots showing the conformational sampling of β-strand

residues during the MD simulation. The MD values are colour-coded from white
to black according to simulation time. Blue regions indicate the allowed and
marginally allowed regions. Red points indicate the starting conformation (one
for each monomer of JIP1-SH3). Panels **b** and **c** demonstrate that the protein
maintains a stable conformation throughout the MD simulation, while
displaying several 180° ring flipping events of Y526 (Fig. 4d).

**Extended Data Table 1 | Data collection and refinement statistics for the JIP1-SH3 structures**

| | JIP1-SH3 Wild type* | JIP1-SH3 H493A* | JIP1-SH3 V517A* | JIP1-SH3 V517L* |
|---|---|---|---|---|
| **Data collection** | | | | |
| Date | 11$^{th}$ Nov 2020 | 14$^{th}$ Nov 2020 | 30$^{th}$ Nov 2020 | 20$^{th}$ Nov 2020 |
| Space group | $P2_1 2_1 2$ | $P2_1$ | $P2_1 2_1 2_1$ | C2 |
| Cell dimensions (Å) | | | | |
| $a, b, c$ (Å) | 71.40, 82.13, 46.85 | 28.05, 45.55, 46.00 | 33.12, 84.25, 98.31 | 78.84, 149.72, 68.09 |
| $\alpha, \beta, \gamma$ (°) | 90.00, 90.00, 90.00 | 90.00, 104.27, 90.00 | 90.00, 90.00, 90.00 | 90.00, 105.83, 90.00 |
| Resolution (Å) | 53.88 – 1.36 | 45.55 – 1.91 | 63.97 – 1.45 | 67.66 – 1.96 |
| | (1.48 – 1.36) † | (2.03 – 1.91) † | (1.52 – 1.45) † | (1.99 – 1.96) † |
| $R_{sym}$ | 0.04 (0.47) | 0.19 (1.50) | 0.06 (0.64) | 0.06 (2.5) |
| $I / \sigma I$ | 9.5 (1.5) | 6.90 (1.18) | 7.0 (1.3) | 8.5 (2.4) |
| Completeness (%) | 91.8 (52.5) | 96.3 (78.0) | 93.9 (47.2) | 98.6 (97.8) |
| Redundancy | 4.6 (5.6) | 6.4 (4.2) | 12.5 (12.9) | 2.7 (2.6) |
| CC (1/2) | 99.3 (61.0) | 99.4 (66.6) | 98.9 (46.4) | 98.8 (92.2) |
| | | | | |
| **Refinement** | | | | |
| Resolution (Å) | 1.45 | 1.95 | 1.45 | 1.96 |
| No. reflections (work/free) | 38196 / 2066 | 7908 / 403 | 34511 / 1815 | 44916 / 2330 |
| $R_{work}$ | 0.159 (0.315) | 0.204 (0.328) | 0.132 (0.159) | 0.224 (0.369) |
| $R_{free}$ | 0.215 (0.376) | 0.253 (0.326) | 0.196 (0.230) | 0.281 (0.367) |
| No. atoms | 2457 | 1087 | 2350 | 6856 |
| Protein | 2094 [A-D] | 1003 [A-B] | 2055 [A-D] | 6219 [A-L] |
| Ions ($PO_4^{3-}$ / $SO_4^{2-}$) | – | – | 5/– | 5 / – |
| Water/PEG | 363 / – | 84 / – | 218/72 | 580 / 52 |
| $B$-factors | | | | |
| Protein | 26.3 | 33.9 | 21.1 | 38.0 |
| Ions ($PO_4^{3-}$ / $SO_4^{2-}$) | – | – | 38.9 / – | 55.5 / – |
| Water/PEG | 37.1 / – | 36.0 / 50.1 | 36.0 / 58.6 | 47.1 / 40.3 |
| R.m.s. deviations | | | | |
| Bond lengths (Å) | 0.013 | 0.010 | 0.012 | 0.011 |
| Bond angles (°) | 1.689 | 1.688 | 1.650 | 1.657 |

*Number of crystals for each structure: 1.

†Values in parenthesis are for the highest-resolution shell.

**Extended Data Table 2 | Data collection and refinement statistics for the JIP1-SH3 and POSH-SH3 structures**

| | JIP1-SH3 A541L* | JIP1-SH3 Y526A* | POSH-SH3-1* | POSH-SH3-4* |
|---|---|---|---|---|
| **Data collection** | | | | |
| Date | 30th Oct 2020 | 5th Dec 2020 | 1st Aug 2019 | 17th Feb 2020 |
| Space group | C2 | $P2_1 2_1 2$ | $P6_5 2 2$ | $P3_2 2 1$ |
| Cell dimensions (Å) | | | | |
| $a, b, c$ (Å) | 96.09, 67.10, 59.14 | 210.64, 62.49, 87.06 | 35.96, 35.96, 181.33 | 45.52, 45.52, 65.74 |
| $\alpha, \beta, \gamma$ (°) | 90.00, 126.70, 90.00 | 90.00, 90.00, 90.00 | 90.00, 90.00, 120.00 | 90.00, 90.00, 120.00 |
| Resolution (Å) | 47.42 – 1.4 | 49.35 – 1.54 | 30.7–1.11 | 65.74–1.45 |
| | (1.51 – 1.40) † | (1.63 – 1.54) † | (1.13–1.11) † | (1.48–1.45) † |
| $R_{sym}$ | 0.081 (0.36) | 0.07 (2.1) | 0.01 (0.4) | 0.01 (0.7) |
| $I / \sigma I$ | 5.5 (1.5) | 11.9 (0.8) | 26.1 (1.4) | 23.8 (0.9) |
| Completeness (%) | 91.9 (52.1) | 99.9 (98.7) | 93.0 (89.9) | 99.9 (99.2) |
| Redundancy | 5.3 (5.0) | 67.7 (6.4) | 25.4 (1.2) | 10.1 (8.5) |
| CC (1/2) | 98.2 (74.0) | 99.9 (30.7) | 0.99 (0.57) | 0.99 (0.52) |
| | | | | |
| **Refinement** | | | | |
| Resolution (Å) | 1.40 | 1.54 | 1.11 | 1.45 |
| No. reflections (work/free) | 39612 / 2116 | 169316 / 1798 | 25335 / 1321 | 13727 / 716 |
| $R_{work}$ | 0.194 (0.328) | 0.138 (0.361) | 0.146 (0.282) | 0.160 (0.489) |
| $R_{free}$ | 0.281 (0.408) | 0.179 (0.379) | 0.177 (0.273) | 0.227 (0.655) |
| No. atoms | 2638 | 7091 | 685 | 637 |
| Protein | 2142 [A-D] | 6129 [A-L] | 531 | 531 |
| Ions ($PO_4^{3-}$ / $SO_4^{2-}$) | – | 10 / 30 | – | – |
| Water/PEG | 522 | 915 | 139 / 15 | 95 / 11 |
| $B$-factors | | | | |
| Protein | 23.6 | 32.8 | 22.3 | 37.5 / 51.4 |
| Ions ($PO_4^{3-}$ / $SO_4^{2-}$) | – | 88.3 / 88.1 | – | – |
| Water/PEG | 41.7 / – | 49.6 / – | 32.1 / 23.6 | 55.9 / 52.6 |
| R.m.s. deviations | | | | |
| Bond lengths (Å) | 0.011 | 0.012 | 0.010 | 0.016 |
| Bond angles (°) | 1.666 | 1.728 | 1.668 | 1.900 |

*Number of crystals for each structure: 1.

†Values in parenthesis are for the highest-resolution shell.

# Reporting Summary

## Statistics

For all statistical analyses, confirm that the following items are present in the figure legend, table legend, main text, or Methods section.

| n/a | Confirmed | |
|---|---|---|
| ☐ | ☒ | The exact sample size (*n*) for each experimental group/condition, given as a discrete number and unit of measurement |
| ☒ | ☐ | A statement on whether measurements were taken from distinct samples or whether the same sample was measured repeatedly |
| ☒ | ☐ | The statistical test(s) used AND whether they are one- or two-sided<br>*Only common tests should be described solely by name; describe more complex techniques in the Methods section.* |
| ☒ | ☐ | A description of all covariates tested |
| ☐ | ☒ | A description of any assumptions or corrections, such as tests of normality and adjustment for multiple comparisons |
| ☐ | ☒ | A full description of the statistical parameters including central tendency (e.g. means) or other basic estimates (e.g. regression coefficient) AND variation (e.g. standard deviation) or associated estimates of uncertainty (e.g. confidence intervals) |
| ☒ | ☐ | For null hypothesis testing, the test statistic (e.g. *F*, *t*, *r*) with confidence intervals, effect sizes, degrees of freedom and *P* value noted<br>*Give P values as exact values whenever suitable.* |
| ☒ | ☐ | For Bayesian analysis, information on the choice of priors and Markov chain Monte Carlo settings |
| ☒ | ☐ | For hierarchical and complex designs, identification of the appropriate level for tests and full reporting of outcomes |
| ☒ | ☐ | Estimates of effect sizes (e.g. Cohen's *d*, Pearson's *r*), indicating how they were calculated |

*Our web collection on statistics for biologists contains articles on many of the points above.*

## Software and code

Policy information about availability of computer code

| Data collection | NMR data acquisition: TopSpin version 3.5, VnmrJ version 3.1<br>X-ray data collection: MXCube version 3 |
|---|---|
| Data analysis | Analysis of NMR relaxation dispersion data: Open source software ChemEx version 2021.4.0-dev2 (https://github.com/gbouvignies/ChemEx)<br>Pocket volume calculations: POVME version 3.0<br>Indexing and integration, X-ray crystallography data: XDS version 20200417, autoProc version 1.1.7<br>Data reduction, X-ray crystallography: Pointless version 1.12.2, Aimless version 0.7.4<br>Molecular replacement, X-ray crystallography: Phaser version 2.8.3<br>Model building and refinement, X-ray crystallography: Coot version 0.8.9.2, Refmac version 5.8.0258, CCP4I version 7.0.078 or 7.1.008 |

For manuscripts utilizing custom algorithms or software that are central to the research but not yet described in published literature, software must be made available to editors and reviewers. We strongly encourage code deposition in a community repository (e.g. GitHub). See the Nature Portfolio guidelines for submitting code & software for further information.

## Data

Policy information about availability of data

All manuscripts must include a data availability statement. This statement should provide the following information, where applicable:

- Accession codes, unique identifiers, or web links for publicly available datasets
- A description of any restrictions on data availability
- For clinical datasets or third party data, please ensure that the statement adheres to our policy

Protein structure data have been deposited in the PDB database with accession codes: 7NYK (JIP1- SH3), 7NZB (JIP1-SH3-V517L), 7NYO (JIP1-SH3-A541L), 7NYL (JIP1-

SH3-H493A), 7NYM (JIP1-SH3-V517A), 7NZC (POSH-SH3-1) and 7NZD (POSH-SH3-4). The 1H, 13C and 15N chemical shifts of JIP1-SH3 have been deposited in the Biological Magnetic Resonance Data Bank with accession codes: 50814 (JIP1-SH3), 50817 (JIP1-SH3-Y526A), 50816 (JIP1-SH3-V517A), 50818 (JIP1-SH3-H493A), 50815 (JIP1-SH3-A541L).
SH3 domain structures for molecular replacement were retrieved from the PDB with accession codes: 2FPE (JIP1), 2LJ1 (Sorbin and SH3 domain-containing protein 1) and 2SRC (tyrosine protein kinase C-Src). Structures for the proteome-wide SH3 sequence analysis were retrieved from the PDB with accession codes: 1CSK, 3A98, 2O9S, 5VEI and 4LNP.

# Field-specific reporting

Please select the one below that is the best fit for your research. If you are not sure, read the appropriate sections before making your selection.

☒ Life sciences  ☐ Behavioural & social sciences  ☐ Ecological, evolutionary & environmental sciences

For a reference copy of the document with all sections, see nature.com/documents/nr-reporting-summary-flat.pdf

# Life sciences study design

All studies must disclose on these points even when the disclosure is negative.

| | |
|---|---|
| Sample size | Statistical methods were not used to determine sample size. |
| Data exclusions | Data were not excluded. |
| Replication | Each crystal structure was solved from a single crystal. All nmr dispersion experiments were performed once with technical replicates as described in the Methods section. |
| Randomization | There was no randomized sample allocation in this work. All tested protein designs received identical treatment. |
| Blinding | Blinding is not relevant for structure determination by X-ray crystallography, blinding was not necessary for other methods used in this study. |

# Reporting for specific materials, systems and methods

We require information from authors about some types of materials, experimental systems and methods used in many studies. Here, indicate whether each material, system or method listed is relevant to your study. If you are not sure if a list item applies to your research, read the appropriate section before selecting a response.

## Materials & experimental systems

| n/a | Involved in the study |
|---|---|
| ☒ | Antibodies |
| ☒ | Eukaryotic cell lines |
| ☒ | Palaeontology and archaeology |
| ☒ | Animals and other organisms |
| ☒ | Human research participants |
| ☒ | Clinical data |
| ☒ | Dual use research of concern |

## Methods

| n/a | Involved in the study |
|---|---|
| ☒ | ChIP-seq |
| ☒ | Flow cytometry |
| ☒ | MRI-based neuroimaging |

