## [Peer Review File · Nature]

Manuscript Title: Visualizing protein breathing motions associated with aromatic ring flipping

Reviewer Comments & Author Rebuttals

Reviewer Reports on the Initial Version:

Referee #1:

It has long been established that aromatic side chains can undergo ring flips, but the details of these structural fluctuations have remained poorly understood. This study employs NMR and X-ray crystallography to obtain insights into large-scale breathing motions required to accommodate ring flips in a buried tyrosine residue in an SH3 domain.

My major concern is that while the study aims to understand a general phenomenon of ring flips seen in many different proteins, the dynamics observed could be very specific to the system studied and may not be generalizable to other proteins. As the authors note, the side chain of Y526 is found in an unusual ground state conformation, and the dynamics seen could be specific to this conformational state. It was not clear to me whether the authors suggest that similar large-scale breathing motions are to be expected in all cases in which ring flips occur. In addition, there is little direct experimental data showing that the excited state is an on-pathway intermediate yet this needs to be established rigorously to support the main claim of the paper. This would require observing two excited states in the NMR data which includes a fully rotated state and a deep kinetic analysis to show that there are no alternative kinetic pathways for ring flipping that bypass the excited state. I did not understand the comment that ring flips seen in MD were orders of magnitude faster than the excited state measured by NMR – this would contradict a role for the excited state as an on-pathway intermediate since in such a case the rates would have to be comparable. I think the authors need to clarify how generalizable their findings are to other ring flips in other proteins (are they always accommodated by the large-scale breathing?) and provide more compelling data and its description for why the excited state is an on-pathway intermediate versus an off-pathway alternative conformation.

Additional comments:

Do NMR spectra of the Y526A mutant show that the mutation does not affect the protein backbone structure? For example are all the chemical shifts as expected purely for the ground state?

I would like to see a direct comparison between the chemical differences measured by NMR CPMG and the difference between the observed chemical shifts for V517A and the WT protein. Do the chemical shifts of V517A match those of the excited state in the WT protein?

Referee #2:

This is a very interesting article, connecting fundamental dynamic processes in proteins, which are made visible by so-called ring flip events (180° jumps of symmetric aromatic amino acids around the χ_2 angle), to a high resolution structural biology view presented in this work. This hopefully informs a broad readership about the dynamic nature of proteins and their conformational plasticity, which is required for their biological function. The high resolution view was possible because another jump event, leading to an alternative state with a 70° moved ring around the χ_2 angle, was identified, and studied in detail with kinetic and structural aspects.

The research in this work is well conducted and the findings very interesting and novel.

Experimental errors are reasonable, and the derived parameters convincing. That's why I support this article to be published, after the following issues are addressed.

Most important, while I am absolutely convinced that the alternative state investigated here, with different orientation of Y526 around χ_2 , shares a lot of the structural features of ring flips (180° jumps around χ_2), since in both the χ_2 angle of the aromatic side chain is altered, I am really not convinced that the structure presented here is an actual kinetic intermediate of the ring flip event.

First, I don't see how a simple and symmetric two-state event like a ring flip would benefit from an asymmetric intermediate in general. For sure it would limit the process to one direction, which would slow the flip down by a factor of two, before being of any benefit.

Second, a lot of ring flip events are really fast per se, ranging from thousands /s to millions /s, and don't seem to need the presence of an intermediate in order to occur. Furthermore, no such intermediates have been found in these studies before, so intermediates are not necessary/beneficial. For very slow ring flip processes, of 10 /s, on the other hand, intermediates and/or even larger structural rearrangements might be needed.

Third and most important, the ring flip of Y526 does appear to be very fast, as judged from MD, with 5 flip events in 1 μ s simulation, pointing to a ring flip kinetics of at least 100,000 /s.

The rate in which the alternative state is formed, however, is just 70 /s, and by this way too slow to matter kinetically (even if the temperature is slightly off or the MD might overestimate the dynamics a bit).

For me it looks like the alternative state described here is kinetically independent from a ring flip, but shares a lot of the structural features (and by this still enables us to learn a lot about ring flips).

It might be that it is slower because the same (maybe) amount of interactions need to be broken, but some of them have to form in a different way in order to form the alternative state, and this is taking longer than just to fall back to the identical ground state.

So, in summary, I think the authors need to put more work in describing differences and similarities of the two kinetic events, the alternative state described here and the ring flip events in other work (and MD in this work). Some visualizations and graphics might also be very nice.

This is especially needed, because of the high impact and reach of this work, with many readers not as familiar with these processes in order to avoid confusion.

The second major point is that very little NMR work is done on the actual side of Y526, which is so central. At least the 5 Tyr signals shown in Ext. Data Fig. 10 should be assigned and quantified in some way, e.g. intensity vs temperature.

Some minor points:

Values described for the NMR-derived kinetics should be rounded to values reflecting the error:
2604 \pm 70 /s should be 2600 \pm 70 /s
2829 \pm 67 /s should be 2830 \pm 70 /s
6756 \pm 294 /s should be 6800 \pm 300 /s

Furthermore, k_{ex} values should also be transferred into the intrinsic rate constants going from the ground state to the alternative state:

wt: k_{12} = 70 /s ; k_{21} = 2530 /s

H493A: $k_{12} = 280 \text{ /s}$; $k_{21} = 2550 \text{ /s}$

This is actually quite interesting, because only the rate from the ground state to the transition state increases, which points to an increased energy of the ground state with the same energy of the transition state and the alternative state, at least in the simplest explanation.

V517A: $k_{12} = 6000 \text{ /s}$; $k_{21} = 740 \text{ /s}$

Can the authors calculate volumes of the three states shown in the movie? Does the increase of volume in their "transition state" compare to reported activation volumes from high-pressure NMR studies?

It appears that region 1 on might behave differently and not be involved in the alternative state. It doesn't change in the Y526A mutant. This should be mentioned.

Can the authors provide information (from the literature or their own experiments) of the thermodynamic stability and folding kinetics of the SH3 domain, or alternatively from other SH3 domains? I think they are usually not that stable. This would allow the reader to compare with ring flips from other proteins, i.e. the very stable BPTI.

I know it is used like this very often, but I still somewhat dislike the phrase excited state, since strictly speaking nothing is excited here (but it is a very exciting state for sure). I think high energy state or alternative state are more accurate. Again this would be a service to the expected broader readership, that are more familiar with UV/Vis or fluorescence.

Referee #3:

A: The ring flip of Y526 in an SH3 domain is investigated that occurs over a rotation about χ_2 of Y526. Normally, and that is also the case here, aromatic rings are packed in such a way in the hydrophobic core of a protein that they cannot simply rotate without affecting the surrounding residues. This is also the case here. The wt shows convincingly relaxation dispersion for a large number of residues which is abolished when Y526 is replaced by alanine. Thus, Y526 is required for observation of the kinetics at the observed time scale of around 2 kHz for the wt and the H493A mutant and almost 7 kHz for the V517A mutant. Due to the fact that wt and the mutants crystallized, one can delineate the conformational path of the rotation of the tyrosine ring in great detail. Very remarkable is the change of the β -sheet from A521 to Y524 that is in a bulge conformation for wt and relaxes to a canonical β -sheet conformation for H493A and V517A (Fig. 4), which exhibit both a staggered χ_2 conformation. To provide from experimental data a "movie" of the reorientation of the phenyl ring is a great achievement.

B: The work is novel and I am not aware of such a clear and impressive study of a phenyl ring rotation. The significance lies in the fact that at room temperature phenyl rings always rotate (they show symmetry in the NMR spectra) while in X-ray structures they appear to be tightly packed between residues of the hydrophobic core surrounding them.

C: The methodology is right and valid. The data are of high quality.

D: Appropriate statistics.

E: Conclusions are justified; they could be strengthened as suggested in the proposal for improvement.

G: References are appropriate.

H: The manuscript is very well written and lucid in all aspects.

F: Suggestions:

1) Very remarkable is the change of the β -sheet from A521 to Y524 that is in a bulge conformation for wt and relaxes to a canonical β -sheet conformation for H493A and V517A (Fig. 4), which exhibit both a staggered χ_2 conformation. It would be nice to know whether this holds true for A541L, which also has a staggered χ_2 conformation (Fig. 3b) but the conformation of the β -sheet is not shown. This should be discussed in the revision.

2) The β -sheet rearrangement specifically involving amino acids 519 and 520 is most remarkable. Indeed it seems that the exchange rates are the largest for Q520 and neighbors. But due to the fact that there are no ticks in Fig. 1a,b, it cannot be really seen well. The ticks should be added, as well as a table where one can see the rates next to the amino acid numbers.

3) In ED Figs 8 and 7 the exchange-dependent R2 rates are shown for the mutants where χ_2 is in the staggered conformation. There the β -sheet is formed. When the ring then flips in these mutants, does the eclipsed conformation of χ_2 then also involve the bulge conformation of the β -sheet?

4) There is an analysis of all SH3 domains with a Y in the position in Fig. 2c. The discovered mechanism of a rotation with a bulge to β -sheet conversion and back is found for the wt of the specific SH3 domain investigated here. I am sure that from the crystal structures in the bioinformatics analysis the authors can extract not only the χ_2 of the tyrosin/phenylalanine ring but also the "status" of the adjacent β -sheet, whether it has a bulge or not. This analysis should be included in order to infer whether the bulge to β -sheet and back mechanism holds for more SH3 domains than the JIP1-SH3 domain.

5) Another point is the following: Due to the fact that the rotation of the phenyl ring can be characterized with a trajectory, chemical shifts could be calculated with empirical programs, maybe even with DFT. How do these variances of chemical shifts taken from the structural trajectory compare with the experimental exchange rates?

6) In Y526A what is the conformation of the β -sheet? In principle it would be energetically lower in the sheet than the bulge conformation. Is that true?

Minor points:

7) χ_2 goes from 0 (eclipsed) to -60 (staggered). Yet it has to go via -120 to -180 the flipped conformation. The -60 and -120 are locally equivalent since for -60 C Δ 2 is eclipsed with H β 2 while for -120 C Δ 1 is eclipsed with H β 1. What conformation do the authors assume for this state?

8) Another point is the following. The ring flip involves rotation about the bond between the sp²-center of C γ and the sp³ center of C β .

"This χ_2 eclipsed conformation is energetically unfavourable and rarely found in proteins due to syn-pentane (steric) interactions between the d1 nuclei of the aromatic ring and the protein backbone (Fig. 2B)".

I would rephrase, since the "staggered" conformation shown in Fig. 2c is also eclipsed between C Δ 2 and H β 2 and rather write that χ_2 is normally adjusted such that C Δ 1 and C Δ 2 seek to be staggered with respect to C α . The syn-pentane interaction depends also on χ_1 .

Author Rebuttals to Initial Comments:

Referee #1:

It has long been established that aromatic side chains can undergo ring flips, but the details of these

structural fluctuations have remained poorly understood. This study employs NMR and X-ray crystallography to obtain insights into large-scale breathing motions required to accommodate ring flips in a buried tyrosine residue in an SH3 domain.

My major concern is that while the study aims to understand a general phenomenon of ring flips seen in many different proteins, the dynamics observed could be very specific to the system studied and may not be generalizable to other proteins. As the authors note, the side chain of Y526 is found in an unusual ground state conformation, and the dynamics seen could be specific to this conformational state. It was not clear to me whether the authors suggest that similar large-scale breathing motions are to be expected in all cases in which ring flips occur.

In addition, there is little direct experimental data showing that the excited state is an on-pathway intermediate yet this needs to be established rigorously to support the main claim of the paper. This would require observing two excited states in the NMR data which includes a fully rotated state and a deep kinetic analysis to show that there are no alternative kinetic pathways for ring flipping that bypass the excited state.

We thank the referee for these comments that have motivated us to rigorously establish whether the observed minor state is an on-pathway intermediate towards full ring flipping. In order to do so, we have acquired new experimental data to determine the ring flipping rate (rotation of 180°) of Y526. Thus, we have:

- 1) Assigned the tyrosine epsilon NMR signals in the ^1H - ^{13}C HSQC of JIP1-SH3.
- 2) Acquired L-optimized TROSY-selected ^{13}C CPMG relaxation dispersion experiments on the side chain of Y526 using a site-selectively $^{13}\text{C}\epsilon$ -labeled sample. This experiment allows to probe processes with exchange rates up to approximately 5000 s^{-1} .
- 3) Acquired on-resonance $^{13}\text{C}\epsilon$ $R_{1\rho}$ relaxation dispersion on the side chain of Y526. This experiment allows to probe processes with exchange rates up to approximately 50000 s^{-1} .

These data and their interpretation have been included in a new, main figure (Fig. 4, replacing previous Fig. 1, which has been moved to Extended Data Fig. 1).

To analyze these new experimental data, we implemented the L-optimized TROSY-selected ^{13}C CPMG dispersion experiment as an analysis module in the program ChemEx. An analysis of these data according to a two-site exchange model yields almost identical exchange parameters compared to the analysis of the $^1\text{H}^{\text{N}}$ and ^{15}N relaxation dispersion data: $k_{\text{EX}} = 2400 \pm 300 \text{ s}^{-1}$ and $p_{\text{minor}} = 2.9 \pm 0.2\%$ (^{13}C) compared to $k_{\text{EX}} = 2600 \pm 70 \text{ s}^{-1}$ and $p_{\text{minor}} = 2.8 \pm 0.1\%$ ($^1\text{H}^{\text{N}}$ and ^{15}N) (Fig. 4b). This provides further insight into the molecular process showing that the contribution from the full ring flipping event to the acquired data is very small or negligible. This strongly suggests that the ring flipping rate is faster than the time scale that can be probed by the CPMG experiment, and that the ring flipping is kinetically independent of the formation of the minor state (assuming that the ^1H and ^{13}C $\epsilon 1$ and $\epsilon 2$ chemical shifts are not degenerate). The contribution to the CPMG curve from the ring flipping therefore amounts to an increase in the “plateau level” at high CPMG frequencies corresponding to a fast exchange contribution to the R_2 rate.

The on-resonance ^{13}C $R_{1\rho}$ relaxation dispersion experiments on the side chain of Y526 can entirely be described by a two-site exchange process with $k_{\text{EX}} = 2400 \text{ s}^{-1}$ and $p_{\text{minor}} = 2.9\%$, as derived from the ^{13}C CPMG relaxation dispersion data (Fig. 4c). No significant modulation of the $R_{1\rho}$ rate at

high spin lock fields was observed demonstrating that ring flipping of Y526 is very fast ($k_{EX} > 50000 \text{ s}^{-1}$).

Although the observed minor state is not an on-pathway intermediate towards full ring flipping, the observed structural rearrangements are required for ring flipping to take place. Specifically, following a suggestion by referee 2, we have analyzed the change in volume of the Y526 pocket during transition from the major (WT conformation) to the minor state (V517A/H493A conformation) (Fig. 4e). Interestingly, we observe an initial cavity expansion (generation of void volume) around the Y526 ring corresponding to 65 \AA^3 that is in excellent agreement with activation volumes derived from pressure-dependent studies of ring flipping rates in other systems ($40\text{-}85 \text{ \AA}^3$). The expansion of the surrounding protein environment is followed by a compaction upon formation of stabilizing CH- π interactions between L519 and Y526 (Fig. 4e, f).

We therefore propose a model, consistent with all experimental and *in silico* data, which links the fast ring flipping with the structural trajectory between the major and minor state. Within this model, fast protein breathing motions along this structural trajectory allow fast ring flipping to take place. These breathing motions mainly involve an opening of the pocket (generation of a void volume) through structural rearrangements of Q520 that is located immediately above the aromatic ring of Y526 in the major state (illustrated in a new Supplementary Video 2). The ring flipping is occasionally interrupted by the formation of stabilizing CH- π interaction via L519 that captures the aromatic ring in a staggered conformation. This formation is accompanied by the β -bulge to β -sheet transition of region 517-522. This stabilization process is rare and occurs on a slow NMR time scale giving rise to the observed CPMG relaxation dispersion.

Interestingly, the same breathing motions, including the initial void volume generation, are observed along the structural trajectory between the WT and the A541L structure (new panel in Extended Data Fig. 6f). Although the A541L structure is not representative of the minor state (CH- π interactions from L519 are prevented in this mutant due to steric clashes with the introduced L541), the structural trajectory is very similar and only differs in the way that the ring is stabilized in the staggered conformation (CH- π interactions from A521 instead of L519). Thus, the same breathing motions, allowing the fast ring flipping, are observed in all three point-mutations (V517A, H493A and A541L).

A summary of the proposed model is illustrated in panel f and g of the new Fig. 4 and is described in the manuscript on page 9-11 under a new section entitled “Void volume creation enables ring flipping”.

As pointed out by the referee, Y526 is found in an unusual ground state conformation, however, it is precisely this fact that allows us to stabilize the minor state by point mutations, as it is associated with an *a priori* more favorable side chain conformation. The unusual ground state conformation should therefore be considered as a tool in this case to probe the breathing motions associated with the ring flipping event.

I did not understand the comment that ring flips seen in MD were orders of magnitude faster than the excited state measured by NMR – this would contradict a role for the excited state as an on-pathway intermediate since in such a case the rates would have to be comparable.

The analysis of our new experimental side chain ^{13}C CPMG and $R_{1\rho}$ relaxation dispersion data establishes a fast ring flipping rate of Y526 ($k_{\text{EX}} > 50000 \text{ s}^{-1}$). The number of ring flips observed during the 1 μs MD simulation (Fig. 4d) is therefore in good agreement with the experimental time scale for ring flipping. The corresponding section in the manuscript has been amended.

I think the authors need to clarify how generalizable their findings are to other ring flips in other proteins (are they always accommodated by the large-scale breathing?) and provide more compelling data and its description for why the excited state is an on-pathway intermediate versus an off-pathway alternative conformation.

Our study suggests that the nature of the breathing motions associated with ring flipping will be different for every aromatic ring as the protein environment will differ in terms of packing density and in terms of the nature of the surrounding residues and the strength of the interactions that they establish with the aromatic ring. The void volume generated along the structural trajectory between the major and minor state, which allows the ring flipping to take place, is in close agreement with previous measurements of ring flipping activation volumes. This shows that the extent of our breathing motions is comparable to that required for ring flipping in other proteins. A recent study suggested that local unfolding of the protein structure would be the source of cavity creation (Dreydoppel et al, JACS 2021), however, our study shows how a substantial void volume can be generated through distinct structural rearrangements, while maintaining the overall protein fold. This is now mentioned in the manuscript on page 11.

In conclusion, the novelty of our manuscript therefore does not lie in the generalizability of the structural rearrangements themselves, but rather in, to our knowledge, the first high-resolution picture of the breathing motions associated with an aromatic ring flipping event. This is a major achievement and resolves a long-standing paradox in the protein field *i.e.* how can aromatic rings undergo ring flipping and pass through the 90° transition state while being surrounded by a tightly packed protein environment.

Additional comments:

Do NMR spectra of the Y526A mutant show that the mutation does not affect the protein backbone structure? For example are all the chemical shifts as expected purely for the ground state?

The Y526A mutation does not affect the protein backbone structure as shown by the high-resolution crystal structure that we determined of this variant (please see Extended Data Figs 3a, c).

I would like to see a direct comparison between the chemical differences measured by NMR CPMG and the difference between the observed chemical shifts for V517A and the WT protein. Do the chemical shifts of V517A match those of the excited state in the WT protein?

The chemical shift difference between the V517A variant and the wild-type protein resembles the CPMG-derived chemical shift difference as shown in Fig. R1_1, however, the agreement is not of the same quality as when we directly compare the CPMG-derived chemical shift differences for the V517A/H493A variants and the wild-type protein (Extended Data Fig. 7f, g and Extended Data Fig. 8f, g). The reason is that the SH3 protein is a small protein domain and a single point mutation affects the chemical shifts of multiple residues. An example of this is the Y526A

mutation for which the backbone conformation is entirely conserved as shown by X-ray crystallography, but that shows chemical shift perturbations as a result of the mutation for almost all backbone $^1\text{H}^{\text{N}}$ and ^{15}N nuclei (Extended Data Fig. 3f, g). We therefore decided to show the agreement between the CPMG-derived chemical shift differences for the two variants (H493A and V517A) with the wild-type protein. In this case, we are comparing chemical shift differences between the major and the minor state within each variant, thereby largely avoiding contributions to the experimental chemical shifts from the mutation sites. This is now explicitly stated in Supplementary Discussion section 2.

Fig. R1_1. Chemical shift difference (absolute value) between the V517A variant and the wild-type protein (blue) compared to the CPMG-derived chemical shift difference between the minor and major state as measured in the wild-type protein (red). Since the V517A variant shows a population of eclipsed versus staggered conformation of 11% versus 89%, the chemical shift differences between the V517A variant and the wild-type protein were extrapolated to a 100% population of the staggered conformation assuming a fast exchange regime.

Referee #2:

This is a very interesting article, connecting fundamental dynamic processes in proteins, which are made visible by so-called ring flip events (180° jumps of symmetric aromatic amino acids around the chi-2 angle), to a high resolution structural biology view presented in this work. This hopefully informs a broad readership about the dynamic nature of proteins and their conformational plasticity, which is required for their biological function. The high resolution view was possible because another jump event, leading to an alternative state with a 70° moved ring around the chi-2 angle, was identified, and studied in detail with kinetic and structural aspects.

The research in this work is well conducted and the findings very interesting and novel. Experimental errors are reasonable, and the derived parameters convincing. That's why I support this article to be published, after the following issues are addressed.

We thank the referee for highlighting the novel aspects of our work and for supporting publication of our manuscript.

Most important, while I am absolutely convinced that the alternative state investigated here, with different orientation of Y526 around chi-2, shares a lot of the structural features of ring flips (180° jumps around chi-2), since in both the chi-2 angle of the aromatic side chain is altered, I am really not convinced that the structure presented here is an actual kinetic intermediate of the ring flip event.

First, I don't see how a simple and symmetric two-state event like a ring flip would benefit from an

asymmetric intermediate in general. For sure it would limit the process to one direction, which would slow the flip down by a factor of two, before being of any benefit.

Second, a lot of ring flip events are really fast per se, ranging from thousands /s to millions /s, and don't seem to need the presence of an intermediate in order to occur. Furthermore, no such intermediates have been found in these studies before, so intermediates are not necessary/beneficial. For very slow ring flip processes, of 10 /s, on the other hand, intermediates and/or even larger structural rearrangements might be needed.

Third and most important, the ring flip of Y526 does appear to be very fast, as judged from MD, with 5 flip events in 1 μ s simulation, pointing to a ring flip kinetics of at least 100,000 /s.

The rate in which the alternative state is formed, however, is just 70 /s, and by this way too slow to matter kinetically (even if the temperature is slightly off or the MD might overestimate the dynamics a bit).

For me it looks like the alternative state described here is kinetically independent from a ring flip, but shares a lot of the structural features (and by this still enables us to learn a lot about ring flips).

It might be that it is slower because the same (maybe) amount of interactions need to be broken, but some of them have to form in a different way in order to form the alternative state, and this is taking longer than just to fall back to the identical ground state.

So, in summary, I think the authors need to put more work in describing differences and similarities of the two kinetic events, the alternative state described here and the ring flip events in other work (and MD in this work). Some visualizations and graphics might also be very nice.

This is especially needed, because of the high impact and reach of this work, with many readers not as familiar with these processes in order to avoid confusion.

The second major point is that very little NMR work is done on the actual side of Y526, which is so central. At least the 5 Tyr signals shown in Ext. Data Fig. 10 should be assigned and quantified in some way, e.g. intensity vs temperature.

We thank the referee for providing such a comprehensive and insightful review of our manuscript. To study the relationship between the observed minor state and full 180° ring flipping, we have carried out new experiments to probe the ring flipping rate of Y526. Thus, we have:

- 1) Assigned the tyrosine epsilon NMR signals in the ^1H - ^{13}C HSQC as suggested by the referee**
- 2) Acquired L-optimized TROSY-selected ^{13}C CPMG relaxation dispersion experiments on the tyrosine side chains using a site-selectively ^{13}C -labeled sample of JIP1-SH3. This experiment allows to probe processes with exchange rates up to approximately 5000 s^{-1} .**
- 3) Acquired on-resonance ^{13}C $R_{1\rho}$ relaxation dispersion on the side chain of Y526. This experiment allows to probe processes with exchange rates up to approximately 50000 s^{-1} .**

These data and their interpretation have been included in a new, main figure (Fig. 4, replacing previous Fig. 1 which has been moved to Extended Data).

To analyze these new experimental data, we implemented the L-optimized TROSY-selected ^{13}C CPMG dispersion experiment as an analysis module in the program ChemEx. An analysis of these data according to a two-site exchange model yields almost identical exchange parameters compared to the analysis of the $^1\text{H}^{\text{N}}$ and ^{15}N relaxation dispersion data: $k_{\text{EX}} = 2400 \pm 300 \text{ s}^{-1}$ and $p_{\text{minor}} = 2.9 \pm 0.2\%$ (^{13}C) compared to $k_{\text{EX}} = 2600 \pm 70 \text{ s}^{-1}$ and $p_{\text{minor}} = 2.8 \pm 0.1\%$ ($^1\text{H}^{\text{N}}$ and ^{15}N) (Fig. 4b). This provides further insight into the molecular process showing that the contribution from the full ring flipping event to the acquired data is very small or negligible. This means that the ring flipping rate is faster than the time scale that can be probed by the CPMG experiment and that the ring flipping is kinetically independent of the formation of the minor state (assuming that the ^1H and ^{13}C ϵ_1 and ϵ_2 chemical shifts are not degenerate). The contribution to the CPMG curve from the ring flipping therefore amounts to an increase in the “plateau level” at high CPMG frequencies corresponding to a fast exchange contribution to the R_2 rate.

The on-resonance ^{13}C $R_{1\rho}$ relaxation dispersion experiments on the side chain of Y526 can entirely be described by a two-site exchange process with $k_{\text{EX}} = 2400 \text{ s}^{-1}$ and $p_{\text{minor}} = 2.9\%$ as derived from the ^{13}C CPMG relaxation dispersion data (Fig. 4c). No significant modulation of the $R_{1\rho}$ rate at high spin lock fields was observed suggesting that ring flipping of Y526 is indeed very fast ($k_{\text{EX}} > 50000 \text{ s}^{-1}$) as pointed out by the referee – an observation that is in agreement with our $1 \mu\text{s}$ MD simulation (Fig. 4d).

Following another suggestion from the referee (see below), we analyzed the change in volume of the Y526 pocket during transition from the major (WT conformation) to the minor state (V517A/H493A conformation) (Fig. 4e). This has been very informative, and we acknowledge the reviewer for this suggestion. Interestingly, we observe an initial cavity expansion (generation of void volume) around the ring corresponding to 65 \AA^3 that is in close agreement with activation volumes derived from pressure-dependent studies of ring flipping rates in other systems ($40\text{--}85 \text{ \AA}^3$). The expansion of the surrounding protein environment is followed by a compaction upon formation of CH- π interactions between L519 and Y526.

We therefore propose a model, consistent with all experimental data, that links the fast ring flipping with the structural trajectory between the major and minor state. In this model, protein breathing motions along this structural trajectory, that are fast on the NMR chemical shift time scale, allow fast ring flipping to take place (illustrated in a new Supplementary Video 2). These breathing motions mainly involve an opening of the pocket (generation of a void volume) through structural rearrangements of Q520 that is located immediately above the aromatic ring of Y526 in the major state. The ring flipping is occasionally interrupted by formation of stabilizing CH- π interaction with L519 that captures the aromatic ring in a staggered conformation. This formation is accompanied by the β -bulge to β -sheet transition of region 517-522. This stabilization process is rare and occurs on a slow NMR time scale giving rise to the observed CPMG relaxation dispersion. The formation of the minor state should therefore be considered as a tool to map out the structural trajectory of the breathing motions. Interestingly, the same breathing motions, including the initial void volume generation, are observed along the structural trajectory between the WT and the A541L structure (new panel in Extended Data Fig. 6f). Although the A541L structure is not representative of the minor state (CH- π interactions from L519 are prevented in this mutant due to steric clashes with the introduced L541), the structural trajectory is very similar and only differs in the way that the ring is stabilized in an intermediate flipping state (CH- π interactions from A521 instead of L519).

The proposed model, uniting all experimental data and *in silico* calculations, is now illustrated in panel f and g of the new Fig. 4 and described in the manuscript on page 9-11 under a new section entitled “Void volume creation enables ring flipping”. We are grateful to the referee for their encouragement to acquire new experimental data and to revise our manuscript thereby better highlighting the differences and similarities between the observed minor state and full ring flipping.

Some minor points:

Values described for the NMR-derived kinetics should be rounded to values reflecting the error:

2604 +/- 70 /s should be 2600 +/- 70 /s
2829 +/- 67 /s should be 2830 +/- 70 /s
6756 +/- 294 /s should be 6800 +/- 300 /s

We agree. This has been corrected.

Furthermore, k_{ex} values should also be transferred into the intrinsic rate constants going from the ground state to the alternative state:

wt: $k_{12} = 70$ /s ; $k_{21} = 2530$ /s
H493A: $k_{12} = 280$ /s ; $k_{21} = 2550$ /s

This is actually quite interesting, because only the rate from the ground state to the transition state increases, which points to an increased energy of the ground state with the same energy of the transition state and the alternative state, at least in the simplest explanation.

V517A: $k_{12} = 6000$ /s ; $k_{21} = 740$ /s

This is an interesting point. We have modified Fig. 3f (previous Fig. 4f) to add the intrinsic rate constants and slightly modified our energy diagrams to reflect, in a more quantitative manner, the energy differences between the major and minor states for the different mutants.

Can the authors calculate volumes of the three states shown in the movie? Does the increase of volume in their "transition state" compare to reported activation volumes from high-pressure NMR studies?

We thank the referee for this excellent suggestion. See above for our comments on this point.

It appears that region 1 on might behave differently and not be involved in the alternative state. It doesn't change in the Y526A mutant. This should be mentioned.

The conformational exchange contributions measured for the WT protein are abolished upon mutation of Y526 to alanine for both ^{15}N and $^1\text{H}^{\text{N}}$ nuclei, including in region 1 (shown in Extended Data Fig. 1d, e and Extended Data Fig. 3d, e). This argues for a role of all three regions (1, 2 and 3) in formation of the minor state.

Can the authors provide information (from the literature or their own experiments) of the thermodynamic stability and folding kinetics of the SH3 domain, or alternatively from other SH3

domains? I think they are usually not that stable. This would allow the reader to compare with ring flips from other proteins, i.e. the very stable BPTI.

We have measured the thermal stability of our SH3 domain using differential scanning calorimetry. It is very stable *i.e.* a measured melting temperature of 82°C. This result is now included in a new panel in Extended Data Figure 1c.

I know it is used like this very often, but I still somewhat dislike the phrase excited state, since strictly speaking nothing is excited here (but it is a very exciting state for sure). I think high energy state or alternative state are more accurate. Again this would be a service to the expected broader readership, that are more familiar with UV/Vis or fluorescence.

We completely agree with the referee on this point. We initially chose to use the term “excited state” to comply with the terminology of previous publications in the NMR field. However, as a service to a broader readership, we have changed “ground state” for “major state” and “excited state” for “minor state” throughout the manuscript. We note that these changes are not highlighted in “track changes” (yellow shading).

Referee #3:

A: The ring flip of Y526 in an SH3 domain is investigated that occurs over a rotation about χ_2 of Y526. Normally, and that is also the case here, aromatic rings are packed in such a way in the hydrophobic core of a protein that they cannot simply rotate without affecting the surrounding residues. This is also the case here. The wt shows convincingly relaxation dispersion for a large number of residues which is abolished when Y526 is replaced by alanine. Thus, Y526 is required for observation of the kinetics at the observed time scale of around 2 kHz for the wt and the H493A mutant and almost 7 kHz for the V517A mutant. Due to the fact that wt and the mutants crystallized, one can delineate the conformational path of the rotation of the tyrosine ring in great detail. Very remarkable is the change of the β -sheet from A521 to Y524 that is in a bulge conformation for wt and relaxes to a canonical β -sheet conformation for H493A and V517A (Fig. 4), which exhibit both a staggered χ_2 conformation. To provide from experimental data a "movie" of the reorientation of the phenyl ring is a great achievement.

We thank the referee for the insightful comments about our work.

B: The work is novel and I am not aware of such a clear and impressive study of a phenyl ring rotation. The significance lies in the fact that at room temperature phenyl rings always rotate (they show symmetry in the NMR spectra) while in X-ray structures they appear to be tightly packed between residues of the hydrophobic core surrounding them.

C: The methodology is right and valid. The data are of high quality.

D: Appropriate statistics.

E: Conclusions are justified; they could be strengthened as suggested in the proposal for improvement.

G: References are appropriate.

H: The manuscript is very well written and lucid in all aspects.

F: Suggestions:

1) Very remarkable is the change of the β -sheet from A521 to Y524 that is in a bulge conformation for wt and relaxes to a canonical β -sheet conformation for H493A and V517A (Fig. 4), which exhibit both a staggered χ_2 conformation. It would be nice to know whether this holds true for A541L, which also has a staggered χ_2 conformation (Fig. 3b) but the conformation of the β -sheet is not shown. This should be discussed in the revision.

The β -sheet conformation of the A541L variant is detailed in the Extended Data Figure 6c-e and referred to in the main text on page 7. As shown in this figure, the β -strand encompassing residues 516-522 loops out leading to the disruption of several hydrogen bonds with the opposing β -strand (encompassing residues 524-529). This region can therefore not be considered as a β -sheet, and it is not possible to characterize the conformation of the A541L variant as “bulge” or “canonical”. This is now mentioned in the legend to Extended Data Figure 6.

2) The β -sheet rearrangement specifically involving amino acids 519 and 520 is most remarkable. Indeed it seems that the exchange rates are the largest for Q520 and neighbors. But due to the fact that there are no ticks in Fig. 1a,b, it cannot be really seen well. The ticks should be added, as well as a table where one can see the rates next to the amino acid numbers.

We suppose that by “exchange rates” the referee means “exchange contributions to the transverse relaxation” that are displayed in Extended Data Fig. 1d, e (note that the previous Fig. 1 has been moved to Extended Data Fig. 1). We have made the ticks on the x-axis clearer, and we have added a table (Supplementary Table 1) with the values of the exchange contributions.

3) In ED Figs 8 and 7 the exchange-dependent R2 rates are shown for the mutants where χ_2 is in the staggered conformation. There the β -sheet is formed. When the ring then flips in these mutants, does the eclipsed conformation of χ_2 then also involve the bulge conformation of the β -sheet?

We apologize if this was not sufficiently clear from the submitted version of the manuscript. In the Extended Data Fig. 7 (for the H493A mutant) and Fig. 8 (for the V517A mutant) we present ^1H and ^{15}N CPMG relaxation dispersion data. These data show that both mutants undergo exchange between two conformations on the micro-to-millisecond time. Analysis of the data demonstrates that the derived chemical shift differences between these two states (for both mutants) closely resemble those of the wild-type protein (Extended Data Fig. 7f, g & 8f, g). This shows that both mutants exchange between the staggered and the eclipsed conformation of Y526 in the same way as the wild-type protein *i.e.* involving the same transition between the canonical β -strand and the β -bulge conformation. This conclusion is also supported by the observed linear dependence of the chemical shifts for the wild-type protein and the two variants (Fig. 3d, e), as explained in the manuscript on page 8.

We have now explicitly stated in the manuscript that the mutants involve the same structural transition between the β -bulge conformation and the canonical β -strand as the wild-type protein (on page 9).

4) There is an analysis of all SH3 domains with a Y in the position in Fig. 2c. The discovered mechanism of a rotation with a bulge to β -sheet conversion and back is found for the wt of the specific SH3 domain investigated here. I am sure that from the crystal structures in the bioinformatics analysis the authors can extract not only the χ_2 of the tyrosin/phenylalanine ring but also the “status” of the adjacent β -sheet, whether it has a bulge or not. This analysis should be included in order to infer whether the bulge to β -sheet and back mechanism holds for more SH3 domains than the JIP1-SH3 domain.

This is an interesting point, and we thank the referee for suggesting this. A detailed comparison shows that the β -bulge conformation at position 518 in JIP1-SH3 is a common feature of all SH3 domains for which high-resolution crystal structures have been solved and for which Y or F is found in position 526 (see Extended Data Figure 5). Our analysis includes the five crystal structures that were previously solved (1CSK, 3A98, 2O9S, 5VEI and 4LNP) and the structures of POSH-SH3-1 and POSH-SH3-4 solved as a part of our work. SH3 domains found in both Group 1 and Group 2 in the principal component analysis (PCA) contain the β -bulge conformation at position 518 suggesting that this specific structural element is not dependent on the side chain conformation of the central tyrosine/phenylalanine and *vice versa*. For this reason, we proposed that the nature of the surrounding residues (position 541, 520, 517 and 493) determines the conformation of the central tyrosine/phenylalanine and that the strength of the CH- π interactions that they mediate with the phenyl moiety is determinant of the stability of the eclipsed versus staggered conformations. This is the basis of the principal component analysis shown in Figure 1c, h, i. It is therefore difficult to predict whether the β -bulge/ β -strand exchange occurs for other SH3 domains and only future case-by-case NMR studies would allow us to reveal such transitions.

In the revised version of the manuscript, we have added a phrase on page 7-8 highlighting that the β -bulge structural element is not determinant of the side chain conformation of the aromatic residue in position 526, and we have modified the Extended Data Figure 5 to include a new panel (panel d) to show that the β -bulge motif is present for all the available crystal structures of the Y/F-class of SH3s.

5) Another point is the following: Due to the fact that the rotation of the phenyl ring can be characterized with a trajectory, chemical shifts could be calculated with empirical programs, maybe even with DFT. How do these variances of chemical shifts taken from the structural trajectory compare with the experimental exchange rates?

According to the referee's suggestion, we have predicted $^1\text{H}^{\text{N}}$ and ^{15}N chemical shifts using the chemical shift prediction tool, SPARTA, for the wild-type protein and the V517A mutant using as input our high-resolution crystal structures. Protons were added to the crystal structures using the program REDUCE prior to chemical shift predictions.

To understand the accuracy by which the chemical shift calculations can be carried out, it is important to remember that $^1\text{H}^{\text{N}}$ and ^{15}N chemical shifts (that we are measuring by the CPMG relaxation dispersion experiments) are in general more difficult to predict than ^{13}C chemical shifts. This is due to their stronger dependence on sample conditions (pH, temperature, buffer composition etc) as well as hydrogen bonding and, in the case of ^1H chemical shifts, also ring current effects. To estimate the expected accuracy for our JIP1-SH3 domain, we initially tested how well $^1\text{H}^{\text{N}}$ and ^{15}N chemical shifts of the major state can be predicted at 25°C using as input the crystal structure of the wild-type SH3 domain (Fig. R3_1).

Fig. R3_1. Difference between experimental and SPARTA-predicted chemical shifts for $^1\text{H}^{\text{N}}$ (left) and ^{15}N (right) of the WT conformation of JIP1-SH3.

We observe a root mean square deviation (RMSD) between the experimental and predicted values of 0.45 ppm for $^1\text{H}^{\text{N}}$ and 2.3 ppm for ^{15}N , in close agreement with the reported RMSDs in the original SPARTA paper on a set of test proteins (similar RMSDs were obtained for another state-of-the-art chemical shift predictor, SHIFTX2). It is therefore important to keep in mind that the RMS prediction error already corresponds to about 50% of the maximum amplitude of the chemical shift differences between the major and minor state ($\Delta\delta_{\text{CPMG}}$) derived from the CPMG relaxation dispersion data (~ 1 ppm for $^1\text{H}^{\text{N}}$ and ~ 5 ppm for ^{15}N , see below).

We next predicted the chemical shifts for the minor state represented by the crystal structure of the V517A mutant. To avoid any influence from the mutation site on the predicted chemical shifts, we re-introduced a valine in position 517 in the structure of the V517A mutant by keeping the original backbone conformation and imposing a side chain conformation of V517 equivalent to the one observed in the H493A crystal structure. We then compared the difference in predicted chemical shifts between the wild-type and the V517A crystal structure to the experimental $\Delta\delta_{\text{CPMG}}$ values (Fig. R3_2).

The agreement between the predicted and experimental chemical shift differences is modest, most likely because SPARTA (and most other available empirical chemical shift predictors) calculates the chemical shifts on the basis of local backbone (ϕ and ψ) and side chain (χ_1) dihedral angles with only a small contribution from through-space interactions. This means that only small chemical shifts changes are predicted in regions, where the structure is invariant between the minor (V517A) and major (wild-type) conformations, while larger shifts are observed experimentally due to the close spatial proximity of these regions to Y526. This is for example the case for region 3 (region surrounding residue 541), where larger chemical shifts changes are observed experimentally compared to prediction.

Due to the limited accuracy with which $^1\text{H}^{\text{N}}$ and ^{15}N chemical shifts can be predicted and due to the fact that empirical chemical shift predictors use local structural information only, we feel that the chemical shift predictions, reported in Fig. R3_2, are not particularly informative and do not provide additional support for the observed structural changes. We therefore feel that these results should not be included as a new figure.

Fig. R3_2. Comparison of experimental and predicted chemical shift differences between the minor and major state of JIP1-SH3 (absolute values). The experimental values (blue) were derived from analysis of the CPMG relaxation dispersion data, while the predicted values (red) were obtained from SPARTA-predictions of chemical shifts using the wild-type and V517A crystal structures as input. Comparisons are shown for both ^{15}N (left) and $^1\text{H}^{\text{N}}$ (right) chemical shifts.

6) In Y526A what is the conformation of the β -sheet? In principle it would be energetically lower in the sheet than the bulge conformation. Is that true?

We apologize if the β -sheet conformation of Y526A was not clear from the manuscript. The Extended Data Fig. 3a shows that the wild-type SH3 and the Y526A variant have near-identical crystal structures. To complement this figure, we have inserted a new panel (panel c) in this figure showing the backbone dihedral angles (ϕ and ψ) for wild-type and Y526A highlighting that the β -bulge conformation is present both in the wild-type protein and in the Y526A mutant.

From our analysis it has become apparent that the β -bulge conformation seems to be the preferred conformation for many SH3 domains (at least of the Y/F-class). Thus, in the context of the SH3 fold, the β -bulge conformation must have generally lower energy compared to the canonical β -sheet conformation.

Minor points:

7) χ_2 goes from 0 (eclipsed) to -60° (staggered). Yet it has to go via -120° to -180° the flipped conformation. The -60° and -120° are locally equivalent since for -60° C α 2 is eclipsed with H β 2 while for -120° C α 1 is eclipsed with H β 1. What conformation do the authors assume for this state?

During the revisions of our manuscript, we have determined the ring flipping rate of Y526 using side chain ^{13}C CPMG and on-resonance $R_{1\rho}$ relaxation dispersion. Our experiments show that ring flipping of Y526 is fast ($k_{\text{EX}} > 50000 \text{ s}^{-1}$). This has allowed us to propose a model directly linking the structural changes between the major (β -bulge) and minor (β -sheet) conformation to the breathing motions necessary for fast ring flipping to take place of Y526. These new experiments and their interpretation are included in a new Fig. 4 (the previous Fig. 1 has been moved to Extended Data Fig. 1 to keep a total of four main figures). Specifically, we show that the structural transition between the major and minor state generates a void volume around the ring of 65 \AA^3 that is in close agreement with measurements of activation volumes for ring flipping in other proteins. The cavity expansion allows fast ring flipping to take place which is occasionally interrupted by formation of stabilizing CH- π interaction with L519 that captures the aromatic ring in a staggered conformation (χ_2 of -60°). This formation is accompanied by the β -bulge to β -sheet transition of region 517-522. This stabilization process is rare and occurs on a slow NMR time scale giving rise to the observed CPMG relaxation dispersion, as illustrated in Fig. 4.

The referee is right that the two conformations corresponding to χ_2 of -60° and -120° are locally equivalent, as defined in the Newman projection. We have no direct structural information on the state corresponding to -120° degrees as it does not appear to be stabilized in the same way as the -60° conformation, probably due to the absence of stabilizing CH- π interactions.

8) Another point is the following. The ring flip involves rotation about the bond between the sp^2 -center of C_γ and the sp^3 center of C_β . “This χ_2 eclipsed conformation is energetically unfavourable and rarely found in proteins due to syn-pentane (steric) interactions between the $d1$ nuclei of the aromatic ring and the protein backbone (Fig. 2B)”. I would rephrase, since the “staggered” conformation shown in Fig. 2c is also eclipsed between $C\Delta 2$ and $H\beta 2$ and rather write that χ_2 is normally adjusted such that $C\Delta 1$ and $C\Delta 2$ seek to be staggered with respect to $C\alpha$. The syn-pentane interaction depends also on χ_1 .

We thank the referee for encouraging us to be more precise in our definition of “staggered” and “eclipsed”. We have corrected the phrase as suggested by the referee (page 5), we have corrected the definition of staggered to specify that it is with respect to $C\delta_1/C\delta_2$ and $C\alpha$ (page 6) and we have modified Fig. 1b to replace the word “syn-pentane interactions” with “steric interactions”.

Reviewer Reports on the First Revision:

Referee #1:

The authors have done a very good job addressing my comments, and the manuscript is now suitable for publication in Nature.

Hashim Al-Hashimi

Referee #2:

The authors have done a great job, have addressed all my points and have done even more very helpful experiments.

The manuscript looks absolutely fine in my opinion, all the errors seem to be appropriate.

Therefore I strongly support the publication of this manuscript.

Referee #3:

All the points raised in my review are adequately addressed in the new version. Therefore I recommend to publish this work as is.